# Regret Minimization in MDPs with Options without Prior Knowledge

**Ronan Fruit**
Sequel Team - Inria Lille
ronan.fruit@inria.fr

**Matteo Pirotta**
Sequel Team - Inria Lille
matteo.pirotta@inria.fr

**Alessandro Lazaric**
Sequel Team - Inria Lille
alessandro.lazaric@inria.fr

**Emma Brunskill**
Stanford University
ebrun@cs.stanford.edu

## Abstract

The option framework integrates temporal abstraction into the reinforcement learning model through the introduction of macro-actions (i.e., options). Recent works leveraged the mapping of Markov decision processes (MDPs) with options to semi-MDPs (SMDPs) and introduced SMDP-versions of exploration-exploitation algorithms (e.g., RMAX-SMDP and UCRL-SMDP) to analyze the impact of options on the learning performance. Nonetheless, the PAC-SMDP sample complexity of RMAX-SMDP can hardly be translated into equivalent PAC-MDP theoretical guarantees, while the regret analysis of UCRL-SMDP requires prior knowledge of the distributions of the cumulative reward and duration of each option, which are hardly available in practice. In this paper, we remove this limitation by combining the SMDP view together with the inner Markov structure of options into a novel algorithm whose regret performance matches UCRL-SMDP's up to an additive regret term. We show scenarios where this term is negligible and the advantage of temporal abstraction is preserved. We also report preliminary empirical results supporting the theoretical findings.

## 1 Introduction

Tractable learning of how to make good decisions in complex domains over many time steps almost definitely requires some form of hierarchical reasoning. One powerful and popular framework for incorporating temporally-extended actions in the context of reinforcement learning is the *options* framework [1]. Creating and leveraging options has been the subject of many papers over the last two decades (see e.g., [2, 3, 4, 5, 6, 7, 8]) and it has been of particular interest recently in combination with deep reinforcement learning, with a number of impressive empirical successes (see e.g., [9] for an application to Minecraft). Intuitively (and empirically) temporal abstraction can help speed up learning (reduce the amount of experience needed to learn a good policy) by shaping the actions selected towards more promising sequences of actions [10], and it can reduce planning computation through reducing the need to evaluate over all possible actions (see e.g., Mann and Mannor [11]). However, incorporating options does not always improve learning efficiency as shown by Jong et al. [12]. Intuitively, limiting action selection only to temporally-extended options might hamper the exploration of the environment by restricting the policy space. Therefore, we argue that in addition to the exciting work being done in heuristic and algorithmic approaches that leverage and/or dynamically discover options, it is important to build a formal understanding of how and when options may help or hurt reinforcement learning performance, and that such insights may also help inform empirically motivated options-RL research.

There has been fairly limited work on formal performance bounds of RL with options. Brunskill and Li [13] derived sample complexity bounds for an RMAX-like exploration-exploitation algorithm for semi-Markov decision processes (SMDPs). While MDPs with options can be mapped to SMDPs, their analysis cannot be immediately translated into the PAC-MDP sample complexity of learning with options, which makes it harder to evaluate their potential benefit. Fruit and Lazaric [14] analyzed an SMDP variant of UCRL [15] showing how its regret can be mapped to the regret of learning in the original MDP with options. The resulting analysis explicitly showed how options can be beneficial whenever the navigability among the states in the original MDP is not compromised (i.e., the MDP diameter is not significantly increased), the level of temporal abstraction is high (i.e., options have long durations, thus reducing the number of decision steps), and the optimal policy with options performs as well as the optimal policy using primitive actions. While this result makes explicit the impact of options on the learning performance, the proposed algorithm (UCRL-SMDP, or SUCRL in short) needs prior knowledge on the parameters of the distributions of cumulative rewards and durations of each option to construct confidence intervals and compute optimistic solutions. In practice this is often a strong requirement and any incorrect parametrization (e.g., loose upper-bounds on the true parameters) directly translates into a poorer regret performance. Furthermore, even if a hand-designed set of options may come with accurate estimates of their parameters, this would not be possible for automatically generated options, which are of increasing interest to the deep RL community. Finally, this prior work views each option as a distinct and atomic macro-action, thus losing the potential benefit of considering the inner structure and the interaction between of options, which could be used to significantly improve sample efficiency.

In this paper we remove the limitations of prior theoretical analyses. In particular, we combine the semi-Markov decision process view on options and the intrinsic MDP structure underlying their execution to achieve temporal abstraction without relying on parameters that are typically unknown. We introduce a transformation mapping each option to an associated irreducible Markov chain and we show that optimistic policies can be computed using only the stationary distributions of the irreducible chains and the SMDP dynamics (i.e., state to state transition probabilities through options). This approach does not need to explicitly estimate cumulative rewards and duration of options and their confidence intervals. We propose two alternative implementations of a general algorithm (FREE-SUCRL, or FSUCRL in short) that differs in whether the stationary distribution of the options' irreducible Markov chains and its confidence intervals are computed explicitly or implicitly through an ad-hoc extended value iteration algorithm. We derive regret bounds for FSUCRL that match the regret of SUCRL up to an additional term accounting for the complexity of estimating the stationary distribution of an irreducible Markov chain starting from its transition matrix. This additional regret is the, possibly unavoidable, cost to pay for not having prior knowledge on options. We further the theoretical findings with a series of simple grid-world experiments where we compare FSUCRL to SUCRL and UCRL (i.e., learning without options).

## 2 Preliminaries

**Learning in MDPs with options.** A finite MDP is a tuple $M = \{\mathcal{S}, \mathcal{A}, p, r\}$ where $\mathcal{S}$ is the set of states, $\mathcal{A}$ is the set of actions, $p(s'|s, a)$ is the probability of transition from state $s$ to state $s'$ through action $a$, $r(s, a)$ is the random reward associated to $(s, a)$ with expectation $\overline{r}(s, a)$. A deterministic policy $\pi : \mathcal{S} \rightarrow \mathcal{A}$ maps states to actions. We define an option as a tuple $o = \{s_o, \beta_o, \pi_o\}$ where $s_o \in \mathcal{S}$ is the state where the option can be initiated[1], $\pi_o : \mathcal{S} \rightarrow \mathcal{A}$ is the associated stationary Markov policy, and $\beta_o : \mathcal{S} \rightarrow [0, 1]$ is the probability of termination. As proved by Sutton et al. [1], when primitive actions are replaced by a set of options $\mathcal{O}$, the resulting decision process is a semi-Markov decision processes (SMDP) $M_{\mathcal{O}} = \{\mathcal{S}_{\mathcal{O}}, \mathcal{O}_s, p_{\mathcal{O}}, R_{\mathcal{O}}, \tau_{\mathcal{O}}\}$ where $\mathcal{S}_{\mathcal{O}} \subseteq \mathcal{S}$ is the set of states where options can start and end, $\mathcal{O}_s$ is the set of options available at state $s$, $p_{\mathcal{O}}(s'|s, o)$ is the probability of terminating in $s'$ when starting $o$ from $s$, $R_{\mathcal{O}}(s, o)$ is the (random) cumulative reward obtained by executing option $o$ from state $s$ until interruption at $s'$ with expectation $\overline{R}_{\mathcal{O}}(s, o)$, and $\tau_{\mathcal{O}}(s, o)$ is the duration (i.e., number of actions executed to go from $s$ to $s'$ by following $\pi_o$) with expectation $\overline{\tau}(s, o)$.[2] Throughout the rest of the paper, we assume that options are well defined.

**Assumption 1.** *The set of options $\mathcal{O}$ is admissible, that is **1)** all options terminate in finite time with probability 1, **2)**, in all possible terminal states there exists at least one option that can start, i.e., $\cup_{o \in \mathcal{O}} \{s : \beta_o(s) > 0\} \subseteq \cup_{o \in \mathcal{O}} \{s_o\}$, **3)** the resulting SMDP $M_{\mathcal{O}}$ is communicating.*

Lem. 3 in [14] shows that under Asm. 1 the family of SMDPs induced by using options in MDPs is such that for any option $o$, the distributions of the cumulative reward and the duration are sub-Exponential with bounded parameters $(\sigma_r(o), b_r(o))$ and $(\sigma_\tau(o), b_\tau(o))$ respectively. The maximal expected duration is denoted by $\tau_{\max} = \max_{s,o} \{\overline{\tau}_{\mathcal{O}}(s, o)\}$. Let $t$ denote primitive action steps and let $i$ index decision steps at option level. The number of decision steps up to (primitive) step $t$ is $N(t) = \max\{n : T_n \leq t\}$, where $T_n = \sum_{i=1}^{n} \tau_i$ is the number of primitive steps executed over $n$ decision steps and $\tau_i$ is the (random) number of steps before the termination of the option chosen at step $i$. Under Asm. 1 there exists a policy $\pi^* : \mathcal{S} \to \mathcal{O}$ over options that achieves the largest gain (per-step reward)

$$\rho_{\mathcal{O}}^* \overset{def}{=} \max_\pi \rho_{\mathcal{O}}^\pi = \max_\pi \lim_{t \to +\infty} \mathbb{E}^\pi \left[ \frac{\sum_{i=1}^{N(t)} R_i}{t} \right], \tag{1}$$

where $R_i$ is the reward cumulated by the option executed at step $i$. The optimal gain also satisfies the optimality equation of an equivalent MDP obtained by data-transformation (Lem. 2 in [16]), i.e.,

$$\forall s \in \mathcal{S} \quad \rho_{\mathcal{O}}^* = \max_{o \in \mathcal{O}_s} \left\{ \frac{\overline{R}_{\mathcal{O}}(s, o)}{\overline{\tau}_{\mathcal{O}}(s, o)} + \frac{1}{\overline{\tau}_{\mathcal{O}}(s, o)} \left( \sum_{s' \in \mathcal{S}} p_{\mathcal{O}}(s'|s, o) u_{\mathcal{O}}^*(s') - u_{\mathcal{O}}^*(s) \right) \right\}, \tag{2}$$

where $u_{\mathcal{O}}^*$ is the optimal bias and $\mathcal{O}_s$ is the set of options than can be started in $s$ (i.e., $o \in \mathcal{O}_s \Leftrightarrow s_o = s$). In the following sections, we drop the dependency on the option set $\mathcal{O}$ from all previous terms whenever clear from the context. Given the optimal average reward $\rho_{\mathcal{O}}^*$, we evaluate the performance of a learning algorithm $\mathfrak{A}$ by its cumulative (SMDP) regret over $n$ decision steps as $\Delta(\mathfrak{A}, n) = \left( \sum_{i=1}^{n} \tau_i \right) \rho_{\mathcal{O}}^* - \sum_{i=1}^{n} R_i$. In [14] it is shown that $\Delta(\mathfrak{A}, n)$ is equal to the MDP regret up to a linear "approximation" regret accounting for the difference between the optimal gains of $M$ on primitive actions and the associated SMDP $M_{\mathcal{O}}$.

## 3 Parameter-free SUCRL for Learning with Options

**Optimism in SUCRL.** At each episode, SUCRL runs a variant of extended value iteration (EVI) [17] to solve the "optimistic" version of the data-transformation optimality equation in Eq. 2, i.e.,

$$\widetilde{\rho}^* = \max_{o \in \mathcal{O}_s} \left\{ \max_{\widetilde{\boldsymbol{R}}, \widetilde{\boldsymbol{\tau}}} \left\{ \frac{\widetilde{R}(s, o)}{\widetilde{\tau}(s, o)} + \frac{1}{\widetilde{\tau}(s, o)} \left( \max_{\widetilde{\boldsymbol{p}}} \left\{ \sum_{s' \in \mathcal{S}} \widetilde{p}(s'|s, o) \widetilde{u}^*(s') \right\} - \widetilde{u}^*(s) \right) \right\} \right\}, \tag{3}$$

where $\widetilde{\boldsymbol{R}}$ and $\widetilde{\boldsymbol{\tau}}$ are the vectors of cumulative rewards and durations for all state-option pairs and they belong to confidence intervals constructed using parameters $(\sigma_r(o), b_r(o))$ and $(\sigma_\tau(o), b_\tau(o))$ (see Sect.3 in [14] for the exact expression). Similarly, confidence intervals need to be computed for $\widetilde{\boldsymbol{p}}$, but this does not require any prior knowledge on the SMDP since the transition probabilities naturally belong to the simplex over states. As a result, without any prior knowledge, such confidence intervals cannot be directly constructed and SUCRL cannot be run. In the following, we see how constructing an irreducible Markov chain (MC) associated to each option avoids this problem.

### 3.1 Irreducible Markov Chains Associated to Options

**Options as absorbing Markov chains.** A natural way to address SUCRL's limitations is to avoid considering options as atomic operations (as in SMDPs) but take into consideration their inner (MDP) structure. Since options terminate in finite time (Asm. 1), they can be seen as an absorbing Markov reward process whose state space contains all states that are reachable by the option and where option terminal states are absorbing states of the MC (see Fig. 1). More formally, for any option $o$ the set of *inner states* $\mathcal{S}_o$ includes the initial state $s_o$ and all states $s$ with $\beta_o(s) < 1$ that are reachable by executing $\pi_o$ from $s_o$ (e.g., $\mathcal{S}_o = \{s_0, s_1\}$ in Fig. 1), while the set of *absorbing states* $\mathcal{S}_o^{\text{abs}}$ includes all states with $\beta_o(s) > 0$ (e.g., $\mathcal{S}_o^{\text{abs}} = \{s_0, s_1, s_2\}$ in Fig. 1). The absorbing MC associated to $o$ is

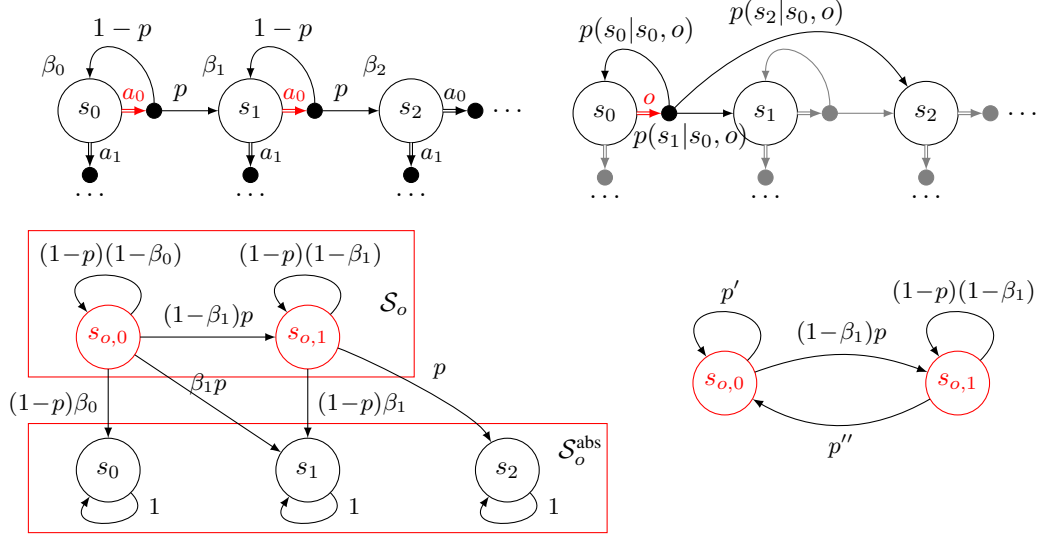

Figure 1: *(upper-left)* MDP with an option $o$ starting from $s_0$ and executing $a_0$ in all states with termination probabilities $\beta_o(s_0) = \beta_0$, $\beta_o(s_1) = \beta_1$ and $\beta_o(s_2) = 1$. *(upper-right)* SMDP dynamics associated to option $o$. *(lower-left)* Absorbing MC associated to options $o$. *(lower-right)* Irreducible MC obtained by transforming the associated absorbing MC with $p' = (1 - \beta_0)(1 - p) + \beta_0(1 - p) + p\beta_1$ and $p'' = \beta_1(1 - p) + p$.

characterized by a transition matrix $P_o$ of dimension $(|\mathcal{S}_o| + |\mathcal{S}_o^{\mathrm{abs}}|) \times (|\mathcal{S}_o| + |\mathcal{S}_o^{\mathrm{abs}}|)$ defined as[3]

$$P_o = \begin{bmatrix} Q_o & V_o \\ 0 & I \end{bmatrix} \text{ with } \begin{aligned} & Q_o(s, s') = (1 - \beta_o(s'))p(s'|s, \pi_o(s)) \text{ for any } s, s' \in \mathcal{S}_o \\ & V_o(s, s') = \beta_o(s')p(s'|s, \pi_o(s)) \text{ for any } s \in \mathcal{S}_o,\ s' \in \mathcal{S}_o^{\mathrm{abs}}, \end{aligned}$$

where $Q_o$ is the transition matrix between inner states (dim. $|\mathcal{S}_o| \times |\mathcal{S}_o|$), $V_o$ is the transition matrix from inner states to absorbing states (dim. $|\mathcal{S}_o| \times |\mathcal{S}_o^{\mathrm{abs}}|$), and $I$ is the identity matrix (dim. $|\mathcal{S}_o^{\mathrm{abs}}| \times |\mathcal{S}_o^{\mathrm{abs}}|$). As proved in Lem. 3 in [14], the expected cumulative rewards $\overline{R}(s, o)$, the duration $\overline{\tau}(s, o)$, and the sub-Exponential parameters $(\sigma_r(o), b_r(o))$ and $(\sigma_\tau(o), b_\tau(o))$ are directly related to the transition matrices $Q_o$ and $V_o$ of the associated absorbing chain $P_o$. This suggests that, given an estimate of $P_o$, we could directly derive the corresponding estimates of $\overline{R}(s, o)$ and $\overline{\tau}(s, o)$. Following this idea, we could "propagate" confidence intervals on the entries of $P_o$ to obtain confidence intervals on rewards and duration estimates without any prior knowledge on their parameters and thus solve Eq. 3 without any prior knowledge. Nonetheless, intervals on $P_o$ do not necessarily translate into compact bounds for $R$ and $\tau$. For example, if the value $\widetilde{V}_o = 0$ belongs to the confidence interval of $\widetilde{P}_o$ (no state in $\mathcal{S}_o^{\mathrm{abs}}$ can be reached), the corresponding optimistic estimates $\widetilde{R}(s, o)$ and $\widetilde{\tau}(s, o)$ are unbounded and Eq. 3 is ill-defined.

**Options as irreducible Markov chains.** We first notice from Eq. 2 that computing the optimal policy only requires computing the ratio $\overline{R}(s, o)/\overline{\tau}(s, o)$ and the inverse $1/\overline{\tau}(s, o)$. Starting from $P_o$, we can construct an irreducible MC whose stationary distribution is directly related to these terms. We proceed as illustrated in Fig. 1: all terminal states are "merged" together and their transitions are "redirected" to the initial state $s_o$. More formally, let $\mathbf{1}$ be the all-one vector of dimension $|\mathcal{S}_o^{\mathrm{abs}}|$, then $v_o = V_o\mathbf{1} \in \mathbb{R}^{|\mathcal{S}_o|}$ contains the cumulative probability to transition from an inner state to any terminal state. Then the chain $P_o$ can be transformed into a MC with transition matrix $P_o' = [v_o \ Q_o'] \in \mathbb{R}^{\mathcal{S}_o \times \mathcal{S}_o}$, where $Q_o'$ contains all but the first column of $Q_o$. $P_o'$ is now an irreducible MC as any state can be reached starting from any other state and thus it admits a unique stationary distribution $\mu_o$. In order to relate $\mu_o$ to the optimality equation in Eq. 2, we need an additional assumption on the options.

**Assumption 2.** *For any option $o \in \mathcal{O}$, the starting state $s_o$ is also a terminal state (i.e., $\beta_o(s_o) = 1$) and any state $s' \in \mathcal{S}$ with $\beta_o(s') < 1$ is an inner state (i.e., $s' \in \mathcal{S}_o$).*

> **Input:** Confidence $\delta \in ]0, 1[$, $r_{\max}$, $\mathcal{S}$, $\mathcal{A}$, $\mathcal{O}$
> **For** episodes $k = 1, 2, ...$ **do**
>
> 1. Set $i_k := i$, $t = t_k$ and episode counters $\nu_k(s, a) = 0$, $\nu_k(s, o) = 0$
> 2. Compute estimates $\widehat{p}_k(s'|s, o)$, $\widehat{P}'_{o,k}$, $\widehat{r}_k(s, a)$ and their confidence intervals in Eq. 6
> 3. Compute an $\epsilon_k$-approximation of the optimal optimistic policy $\widetilde{\pi}_k$ of Eq. 5
> 4. **While** $\forall l \in [t + 1, t + \tau_i]$, $\nu_k(s_l, a_l) < N_k(s_l, a_l)$ **do**
>    (a) Execute option $o_i = \widetilde{\pi}_k(s_i)$, obtain primitive rewards $r_i^1, ..., r_i^{\tau_i}$ and visited states $s_i^1, ..., s_i^{\tau_i} = s_{i+1}$
>    (b) Set $\nu_k(s_i, o_i) \mathrel{+}= 1$, $i \mathrel{+}= 1$, $t \mathrel{+}= \tau_i$ and $\nu_k(s, \pi_{o_i}(s)) \mathrel{+}= 1$ for all $s \in \{s_i^1, ..., s_i^{\tau_i}\}$
> 5. Set $N_k(s, o) \mathrel{+}= \nu_k(s, o)$ and $N_k(s, a) \mathrel{+}= \nu_k(s, a)$

Figure 2: The general structure of FSUCRL.

While the first part has a very minor impact on the definition of $\mathcal{O}$, the second part of the assumption guarantees that options are "well designed" as it requires the termination condition to be coherent with the *true* inner states of the option, so that if $\beta_o(s') < 1$ then $s'$ should be indeed reachable by the option. Further discussion about Asm. 2 is reported in App. A. We then obtain the following property.

**Lemma 1.** *Under Asm. 2, let $\boldsymbol{\mu_o} \in [0, 1]^{\mathcal{S}_o}$ be the unique stationary distribution of the irreducible MC $P'_o$ associated to option o, then* [4]

$$\forall s \in \mathcal{S}, \ \forall o \in \mathcal{O}_s, \quad \frac{1}{\overline{\tau}(s, o)} = \mu_o(s) \quad \text{and} \quad \frac{\overline{R}(s, o)}{\overline{\tau}(s, o)} = \sum_{s' \in \mathcal{S}_o} \overline{r}(s', \pi_o(s'))\mu_o(s'). \quad (4)$$

This lemma illustrates the relationship between the stationary distribution of $P'_o$ and the key terms in Eq. 2.[5] As a result, we can apply Lem. 1 to Eq. 3 and obtain the optimistic optimality equation

$$\forall s \in \mathcal{S} \quad \widetilde{\rho}^* = \max_{o \in \mathcal{O}_s} \left\{ \max_{\widetilde{\boldsymbol{\mu}}_o, \widetilde{\boldsymbol{r}}_o} \left\{ \sum_{s' \in \mathcal{S}_o} \widetilde{r}_o(s')\,\widetilde{\mu}_o(s') + \widetilde{\mu}_o(s)\Big( \max_{\widetilde{\boldsymbol{b}}_o} \{\widetilde{\boldsymbol{b}}_o^{\mathsf{T}} \widetilde{\boldsymbol{u}}^*\} - \widetilde{u}^*(s) \Big) \right\} \right\}, \quad (5)$$

where $\widetilde{r}_o(s') = \widetilde{r}(s', \pi_o(s'))$ and $\widetilde{\boldsymbol{b}}_o = (\widetilde{p}(s'|s, o))_{s' \in \mathcal{S}}$. Unlike in the absorbing MC case, where compact confidence sets for $P_o$ may lead to unbounded optimistic estimates for $\widetilde{R}$ and $\widetilde{\tau}$, in this formulation $\mu_o(s)$ can be equal to 0 (i.e., infinite duration and cumulative reward) without compromising the solution of Eq. 5. Furthermore, estimating $\mu_o$ implicitly leverages over the correlation between cumulative reward and duration, which is ignored when estimating $\overline{R}(s, o)$ and $\overline{\tau}(s, o)$ separately. Finally, we prove the following result.

**Lemma 2.** *Let $\widetilde{r}_o \in \mathcal{R}$, $\widetilde{\boldsymbol{b}}_o \in \mathcal{P}$, and $\widetilde{\mu}_o \in \mathcal{M}$, with $\mathcal{R}$, $\mathcal{P}$, $\mathcal{M}$ compact sets containing the true parameters $\overline{r}_o$, $\boldsymbol{b}_o$ and $\mu_o$, then the optimality equation in Eq. 5 always admits a unique solution $\widetilde{\rho}^*$ and $\widetilde{\rho}^* \geq \rho^*$ (i.e., the solution of Eq. 5 is an optimistic gain).*

Now, we need to provide an explicit algorithm to compute the optimistic optimal gain $\widetilde{\rho}^*$ of Eq. 5 and its associated optimistic policy. In the next section, we introduce two alternative algorithms that are guaranteed to compute an $\epsilon$-optimistic policy.

### 3.2 SUCRL with Irreducible Markov Chains

The structure of the UCRL-like algorithm for learning with options but with no prior knowledge on distribution parameters (called FREE-SUCRL, or FSUCRL) is reported in Fig. 2. Unlike SUCRL we do not directly estimate the expected cumulative reward and duration of options but we estimate the SMDP transition probabilities $p(s'|s, o)$, the irreducible MC $P'_o$ associated to each option, and the state-action reward $\overline{r}(s, a)$. For all these terms we can compute confidence intervals (Hoeffding and empirical Bernstein) without any prior knowledge as

$$\left| r(s,a) - \widehat{r}_k(s,a) \right| \le \beta_k^r(s,a) \propto r_{\max} \sqrt{\frac{\log(SAt_k/\delta)}{N_k(s,a)}}, \tag{6a}$$

$$\left| p(s'|s,o) - \widehat{p}_k(s'|s,o) \right| \le \beta_k^p(s,o,s') \propto \sqrt{\frac{2\widehat{p}_k(s'|s,o)\left(1 - \widehat{p}_k(s'|s,o)\right)c_{t_k,\delta}}{N_k(s,o)}} + \frac{7c_{t_k,\delta}}{3N_k(s,o)}, \tag{6b}$$

$$\left| P'_o(s,s') - \widehat{P}'_{o,k}(s,s') \right| \le \beta_k^P(s,o,s') \propto \sqrt{\frac{2\widehat{P}'_{o,k}(s,s')\left(1 - \widehat{P}'_{o,k}(s,s')\right)d_{t_k,\delta}}{N_k(s,\pi_o(s))}} + \frac{7d_{t_k,\delta}}{3N_k(s,\pi_o(s))}, \tag{6c}$$

where $N_k(s,a)$ (resp. $N_k(s,o)$) is the number of samples collected at state-action $s,a$ (resp. state-option $s,o$) up to episode $k$, Eq. 6a coincides with the one used in UCRL, in Eq. 6b $s = s_o$ and $s' \in \mathcal{S}$, and in Eq. 6c $s,s' \in \mathcal{S}_o$. Finally, we set $c_{t_k,\delta} = O\left(\log\left(SOt_k\right)/\delta\right)$ and $d_{t_k,\delta} = O\left(\log\left(|\mathcal{S}_o|\log(t_k)/\delta\right)\right)$ [18, Eq. 31].

To obtain an actual implementation of the algorithm reported on Fig. 2 we need to define a procedure to compute an approximation of Eq. 5 (step 3). Similar to UCRL and SUCRL, we define an EVI algorithm starting from a function $u_0(s) = 0$ and computing at each iteration $j$

$$u_{j+1}(s) = \max_{o \in \mathcal{O}_s}\left\{ \max_{\widetilde{\boldsymbol{\mu}_o}}\left\{ \sum_{s' \in \mathcal{S}_o} \widetilde{r}_o(s')\,\widetilde{\mu}_o(s') + \widetilde{\mu}_o(s)\left( \max_{\widetilde{\boldsymbol{b}_o}}\left\{ \widetilde{\boldsymbol{b}_o^{\mathsf{T}}\boldsymbol{u_j}} \right\} - u_j(s) \right) \right\} \right\} + u_j(s), \tag{7}$$

where $\widetilde{r}_o(s')$ is the optimistic reward (i.e., estimate plus the confidence bound of Eq. 6a) and the optimistic transition probability vector $\widetilde{\boldsymbol{b}_o}$ is computed using the algorithm introduced in [19, App. A] for Bernstein bound as in Eqs. 6b, 6c or in [15, Fig. 2] for Hoeffding bound (see App. B).

Depending on whether confidence intervals for $\mu_o$ are computed explicitly or implicitly we can define two alternative implementations that we present below.

**Explicit confidence intervals.** Given the estimate $\widehat{P}'_o$, let $\widehat{\mu}_o$ be the solution of $\widehat{\boldsymbol{\mu}_o^{\mathsf{T}}} = \widehat{\boldsymbol{\mu}_o^{\mathsf{T}}}\widehat{P}'_o$ under constraint $\widehat{\boldsymbol{\mu}_o^{\mathsf{T}}}e = e$. Such a $\widehat{\boldsymbol{\mu}_o}$ always exists and is unique since $\widehat{P}'_o$ is computed after terminating the option at least once and is thus irreducible. The perturbation analysis in [20] can be applied to derive the confidence interval

$$\|\boldsymbol{\mu_o} - \widehat{\boldsymbol{\mu}_o}\|_1 \le \beta_k^\mu(o) := \widehat{\kappa}_{o,\min}\|P'_o - \widehat{P}'_o\|_{\infty,1}, \tag{8}$$

where $\|\cdot\|_{\infty,1}$ is the maximum of the $\ell_1$-norm of the rows of the transition matrix, $\widehat{\kappa}_{o,\min}$ is the smallest condition number[6] for the $\ell_1$-norm of $\boldsymbol{\mu_o}$. Let $\boldsymbol{\zeta_o} \in \mathbb{R}^{|\mathcal{S}_o|}$ be such that $\zeta_o(s_o) = \widetilde{r}_o(s_o) + \max_{\widetilde{\boldsymbol{b}_o}}\{\widetilde{\boldsymbol{b}_o^{\mathsf{T}}\boldsymbol{u_j}}\} - u_j(s_o)$ and $\zeta_o(s) = \widetilde{r}_o(s)$, then the maximum over $\widetilde{\boldsymbol{\mu}}_o$ in Eq. 7 has the same form as the innermost maximum over $\boldsymbol{b}_o$ (with Hoeffding bound) and thus we can directly apply Alg. [15, Fig. 2] with parameters $\widehat{\boldsymbol{\mu}}_o, \beta_k^\mu(o)$, and states $\mathcal{S}_o$ ordered descendingly according to $\boldsymbol{\zeta}_o$. The resulting value is then directly plugged into Eq. 7 and $u_{j+1}$ is computed. We refer to this algorithm as FSUCRLv1.

**Nested extended value iteration.** An alternative approach builds on the observation that the maximum over $\boldsymbol{\mu}_o$ in Eq. 7 can be seen as the optimization of the average reward (gain)

$$\widetilde{\rho}_o^*(u_j) = \max_{\widetilde{\boldsymbol{\mu}_o}}\left\{ \sum_{s' \in \mathcal{S}_o} \zeta_o(s')\widetilde{\mu}_o(s') \right\}, \tag{9}$$

where $\zeta_o$ is defined as above. Eq. 9 is indeed the optimal gain of a bounded-parameter MDP with state space $\mathcal{S}_o$, an action space composed of the option action (i.e., $\pi_o(s)$), and transitions $\widetilde{P}'_o$ in the confidence intervals [7] of Eq. 6c, and thus we can write its optimality equation

$$\widetilde{\rho}_o^*(u_j) = \max_{\widetilde{P}'_o}\left\{ \zeta_o(s) + \sum_{s'} \widetilde{P}'_o(s,s')\widetilde{w}_o^*(s') \right\} - \widetilde{w}_o^*(s), \tag{10}$$

where $\widetilde{w}_o^*$ is an optimal bias. For any input function $v$ we can compute $\rho_o^*(v)$ by using EVI on the bounded-parameter MDP, thus avoiding to explicitly construct the confidence intervals of $\widetilde{\mu}_o$. As a result, we obtain two nested EVI algorithms where, starting from an initial bias function $v_0(s) = 0$,[8] at any iteration $j$ we set the bias function of the inner EVI to $w_{j,0}^o(s) = 0$ and we compute (see App. C.3 for the general EVI for bounded-parameter MDPs and its guarantees)

$$w_{j,l+1}^o(s') = \max_{\widetilde{P}_o} \left\{ \zeta_o(s) + \widetilde{\boldsymbol{P}_o}(\cdot|\boldsymbol{s'})^\mathsf{T} \boldsymbol{w}_{\boldsymbol{j,l}}^o \right\}, \tag{11}$$

until the stopping condition $l_j^o = \inf\{l \geq 0 : \mathrm{sp}\{\boldsymbol{w_{j,l+1}^o} - \boldsymbol{w_{j,l}^o}\} \leq \varepsilon_j\}$ is met, where $(\varepsilon_j)_{j\geq 0}$ is a vanishing sequence. As $w_{j,l+1}^o - w_{j,l}^o$ converges to $\rho_o^*(v_j)$ with $l$, the outer EVI becomes

$$v_{j+1}(s) = \max_{o \in \mathcal{O}_s} \left\{ g\big(\boldsymbol{w_{j,l_j^o+1}^o} - \boldsymbol{w_{j,l_j^o}^o}\big) \right\} + v_j(s), \tag{12}$$

where $g : \boldsymbol{v} \mapsto \frac{1}{2} (\max\{\boldsymbol{v}\} + \min\{\boldsymbol{v}\})$. In App. C.4 we show that this nested scheme, that we call FSUCRLv2, converges to the solution of Eq. 5. Furthermore, if the algorithm is stopped when $\mathrm{sp}\{\boldsymbol{v_{j+1}} - \boldsymbol{v_j}\} + \varepsilon_j \leq \varepsilon$ then $|\widehat{\rho}^* - g(\boldsymbol{v_{j+1}} - \boldsymbol{v_j})| \leq \varepsilon/2$.

One of the interesting features of this algorithm is its hierarchical structure. Nested EVI is operating on two different time scales by iteratively considering every option as an independent optimistic planning sub-problem (EVI of Eq. 11) and gathering all the results into a higher level planning problem (EVI of Eq. 12). This idea is at the core of the hierarchical approach in RL, but it is not always present in the algorithmic structure, while nested EVI naturally arises from decomposing Eq. 7 in two value iteration algorithms. It is also worth to underline that the confidence intervals implicitly generated for $\widetilde{\mu}_o$ are never worse than those in Eq. 8 and they are often much tighter. In practice the bound of Eq. 8 may be actually worse because of the worst-case scenario considered in the computation of the condition numbers (see Sec. 5 and App. F).

## 4 Theoretical Analysis

Before stating the guarantees for FSUCRL, we recall the definition of diameter of $M$ and $M_\mathcal{O}$:

$$D = \max_{s,s' \in \mathcal{S}} \min_{\pi:\mathcal{S}\to\mathcal{A}} \mathbb{E}\big[\tau_\pi(s,s')\big], \quad D_\mathcal{O} = \max_{s,s' \in \mathcal{S}_\mathcal{O}} \min_{\pi:\mathcal{S}\to\mathcal{O}} \mathbb{E}\big[\tau_\pi(s,s')\big],$$

where $\tau_\pi(s,s')$ is the (random) number of primitive actions to move from $s$ to $s'$ following policy $\pi$. We also define a *pseudo-diameter* characterizing the "complexity" of the inner dynamics of options:

$$\widetilde{D}_\mathcal{O} = \frac{r^* \kappa_*^1 + \tau_{\max} \kappa_*^\infty}{\sqrt{\mu^*}}$$

where we define:

$$r^* = \max_{o \in \mathcal{O}} \{\mathrm{sp}(r_o)\}, \quad \kappa_*^1 = \max_{o \in \mathcal{O}} \{\kappa_o^1\}, \quad \kappa_*^\infty = \max_{o \in \mathcal{O}} \{\kappa_o^\infty\}, \quad \text{and } \mu^* = \min_{o \in \mathcal{O}} \left\{ \min_{s \in \mathcal{S}_o} \mu_o(s) \right\}$$

with $\kappa_o^1$ and $\kappa_o^\infty$ the condition numbers of the irreducible MC associated to options $o$ (for the $\ell_1$ and $\ell_\infty$-norm respectively [20]) and $\mathrm{sp}(r_o)$ the span of the reward of the option. In App. D we prove the following regret bound.

**Theorem 1.** *Let $M$ be a communicating MDP with reward bounded between 0 and $r_{\max} = 1$ and let $\mathcal{O}$ be a set of options satisfying Asm. 1 and 2 such that $\sigma_r(s,o) \leq \sigma_r$, $\sigma_\tau(s,o) \leq \sigma_\tau$, and $\overline{\tau}(s,o) \leq \tau_{\max}$. We also define $B_\mathcal{O} = \max_{s,o} supp(p(\cdot|s,o))$ (resp. $B = \max_{s,a} supp(p(\cdot|s,a))$ as the largest support of the SMDP (resp. MDP) dynamics. Let $T_n$ be the number of primitive steps executed when running FSUCRLv2 over $n$ decision steps, then its regret is bounded as*

$$\Delta(\mathrm{FSUCRL}, n) = \widetilde{O}\bigg( \underbrace{D_\mathcal{O}\sqrt{SB_\mathcal{O}On}}_{\Delta_p} + \underbrace{(\sigma_r + \sigma_\tau)\sqrt{n}}_{\Delta_{R,\tau}} + \underbrace{\sqrt{SAT_n} + \widetilde{D}_\mathcal{O}\sqrt{SBOT_n}}_{\Delta_\mu} \bigg) \tag{13}$$

**Comparison to SUCRL.** Using the confidence intervals of Eq. 6b and a slightly tighter analysis than the one by Fruit and Lazaric [14] (Bernstein bounds and higher accuracy for EVI) leads to a regret bound for SUCRL as

$$\Delta(\text{SUCRL}, n) = \widetilde{O}\Big(\Delta_p + \Delta_{R,\tau} + \underbrace{(\sigma_r^+ + \sigma_\tau^+)\sqrt{SAn}}_{\Delta'_{R,\tau}}\Big), \tag{14}$$

where $\sigma_r^+$ and $\sigma_\tau^+$ are upper-bounds on $\sigma_r$ and $\sigma_\tau$ that are used in defining the confidence intervals for $\tau$ and $R$ that are actually used in SUCRL. The term $\Delta_p$ is the regret induced by errors in estimating the SMDP dynamics $p(s'|s,o)$, while $\Delta_{R,\tau}$ summarizes the randomness in the cumulative reward and duration of options. Both these terms scale as $\sqrt{n}$, thus taking advantage of the temporal abstraction (i.e., the ratio between the number of primitive steps $T_n$ and the decision steps $n$). The main difference between the two bounds is then in the last term, which accounts for the regret due to the optimistic estimation of the behavior of the options. In SUCRL this regret is linked to the upper bounds on the parameters of $R$ and $\tau$. As shown in Thm.2 in [14], when $\sigma_r^+ = \sigma_r$ and $\sigma_\tau^+ = \sigma_\tau$, the bound of SUCRL is nearly-optimal as it almost matches the lower-bound, thus showing that $\Delta'_{R,\tau}$ is unavoidable. In FSUCRL however, the additional regret $\Delta_\mu$ comes from the estimation errors of the per-time-step rewards $r_o$ and the dynamic $P'_o$. Similar to $\Delta_p$, these errors are amplified by the pseudo-diameter $\widetilde{D}_\mathcal{O}$. While $\Delta_\mu$ may actually be the unavoidable cost to pay for removing the prior knowledge about options, it is interesting to analyze how $\widetilde{D}_\mathcal{O}$ changes with the structure of the options (see App. E for a concrete example). The probability $\mu_o(s)$ decreases as the probability of visiting an inner state $s \in \mathcal{S}_o$ using the option policy. In this case, the probability of collecting samples on the inner transitions is low and this leads to large estimation errors for $P'_o$. These errors are then propagated to the stationary distribution $\mu_o$ through the condition numbers $\kappa$ (e.g., $\kappa_o^1$ directly follows from an non-empirical version of Eq. 8). Furthermore, we notice that $1/\mu_o(s) \geq \tau_o(s) \geq |\mathcal{S}_o|$, suggesting that "long" or "big" options are indeed more difficult to estimate. On the other hand, $\Delta_\mu$ becomes smaller whenever the transition probabilities under policy $\pi_o$ are supported over a few states ($B$ small) and the rewards are similar within the option ($\text{sp}(r_o)$ small). While in the worst case $\Delta_\mu$ may actually be much bigger than $\Delta'_{R,\tau}$ when the parameters of $R$ and $\tau$ are accurately known (i.e., $\sigma_\tau^+ \approx \sigma_\tau$ and $\sigma_r^+ \approx \sigma_r$), in Sect. 5 we show scenarios in which the actual performance of FSUCRL is close or better than SUCRL and the advantage of learning with options is preserved.

To explain why FSUCRL can perform better than SUCRL we point out that FSUCRL's bound is somewhat worst-case w.r.t. the correlation between options. In fact, in Eq. 6c the error in estimating $P'_o$ in a state $s$ does not scale with the number of samples obtained while executing option $o$ but those collected by taking the primitive action prescribed by $\pi_o$. This means that even if $o$ has a low probability of reaching $s$ starting from $s_o$ (i.e., $\mu_o(s)$ is very small), the *true* error may still be small as soon as another option $o'$ executes the same action (i.e., $\pi_o(s) = \pi_{o'}(s)$). In this case the regret bound is loose and the actual performance of FSUCRL is much better. Therefore, although it is not apparent in the regret analysis, not only is FSUCRL leveraging on the correlation between the cumulative reward and duration of a single option, but it is also leveraging on the correlation between different options that share inner state-action pairs.

**Comparison to UCRL.** We recall that the regret of UCRL is bounded as $O(D\sqrt{SBAT_n})$, where $T_n$ is to the total number of steps. As discussed by [14], the major advantage of options is in terms of temporal abstraction (i.e., $T_n \gg n$) and reduction of the state-action space (i.e., $\mathcal{S}_\mathcal{O} < \mathcal{S}$ and $O < A$). Eq.(13) also reveals that options can also improve the learning speed by reducing the size of the support $B_\mathcal{O}$ of the dynamics of the environment w.r.t. primitive actions. This can lead to a huge improvement e.g., when options are designed so as to reach a specific goal. This potential advantage is new compared to [14] and matches the intuition on "good" options often presented in the literature (see e.g., the concept of "funnel" actions introduced by Dietterich [23]).

**Bound for FSUCRLv1.** Bounding the regret of FSUCRLv1 requires bounding the empirical $\widehat{\kappa}$ in Eq. (8) with the true condition number $\kappa$. Since $\widehat{\kappa}$ tends to $\kappa$ as the number of samples of the option increases, the overall regret would only be increased by a lower order term. In practice however, FSUCRLv2 is preferable to FSUCRLv1. The latter will suffer from the true condition numbers $(\kappa_o^1)_{o \in \mathcal{O}}$ since they are used to compute the confidence bounds on the stationary distributions $(\mu_o)_{o \in \mathcal{O}}$, while for FSUCRLv2 they appear only in the analysis. As much as the dependency on the diameter in the analysis of UCRL, the condition numbers may also be loose in practice, although tight from a theoretical perspective. See App.D.6 and experiments for further insights.

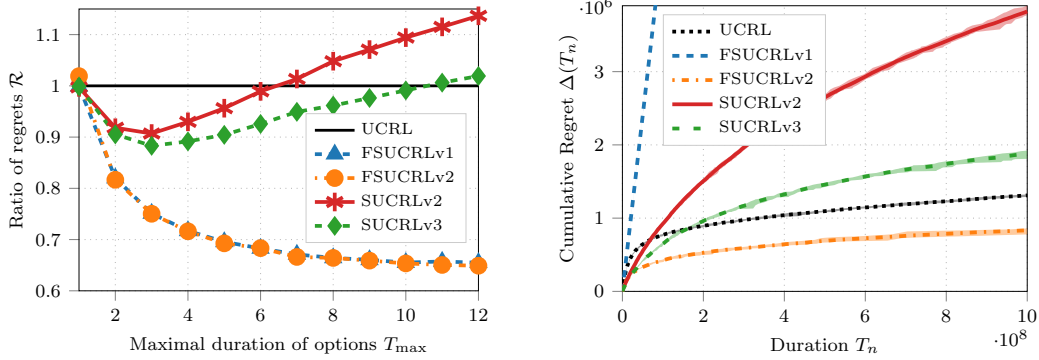

Figure 3: *(Left)* Regret after $1.2 \cdot 10^8$ steps normalized w.r.t. UCRL for different option durations in a 20x20 grid-world. *(Right)* Evolution of the regret as $T_n$ increases for a 14x14 four-rooms maze.

## 5  Numerical Simulations

In this section we compare the regret of FSUCRL to SUCRL and UCRL to empirically verify the impact of removing prior knowledge about options and estimating their structure through the irreducible MC transformation. We consider the toy domain presented in [14] that was specifically designed to show the advantage of temporal abstraction and the classical 4-rooms maze [1]. To be able to reproduce the results of [14], we run our algorithm with Hoeffding confidence bounds for the $\ell_1$-deviation of the empirical distribution (implying that $B_{\mathcal{O}}$ has no impact). We consider settings where $\Delta_{R,\tau}$ is the dominating term of the regret (refer to App. F for details).

When comparing the two versions of FSUCRL to UCRL on the grid domain (see Fig. 3 *(left)*), we empirically observe that the advantage of temporal abstraction is indeed preserved when removing the knowledge of the parameters of the option. This shows that the benefit of temporal abstraction is not just a mere artifact of prior knowledge on the options. Although the theoretical bound in Thm. 1 is always worse than its SMDP counterpart (14), we see that FSUCRL performs much better than SUCRL in our examples. This can be explained by the fact that the options we use greatly overlap. Even if our regret bound does not make explicit the fact that FSUCRL exploits the correlation between options, this can actually significantly impact the result in practice. The two versions of SUCRL differ in the amount of prior knowledge given to the algorithm to construct the parameters $\sigma_r^+$ and $\sigma_\tau^+$ that are used in building the confidence intervals. In *v3* we provide a tight upper-bound $r_{\max}$ on the rewards and distinct option-dependent parameters for the duration ($\tau_o$ and $\sigma_\tau(o)$), in *v2* we only provide a global (option-independent) upper bound on $\tau_o$ and $\sigma_o$. Unlike FSUCRL which is "parameter-free", SUCRL is highly sensitive to the prior knowledge about options and can perform even worse than UCRL. A similar behaviour is observed in Fig. 3 *(right)* where both the versions of SUCRL fail to beat UCRL but FSUCRLv2 has nearly half the regret of UCRL. On the contrary, FSUCRLv1 suffers a linear regret due to a loose dependency on the condition numbers (see App. F.2). This shows that the condition numbers appearing in the bound of FSUCRLv2 are actually loose. In both experiments, UCRL and FSUCRL had similar running times meaning that the improvement in cumulative regret is not at the expense of the computational complexity.

## 6  Conclusions

We introduced FSUCRL, a parameter-free algorithm to learn in MDPs with options by combining the SMDP view to estimate the transition probabilities at the level of options ($p(s'|s,o)$) and the MDP structure of options to estimate the stationary distribution of an associated irreducible MC which allows to compute the optimistic policy at each episode. The resulting regret matches SUCRL bound up to an additive term. While in general, this additional regret may be large, we show both theoretically and empirically that FSUCRL is actually competitive with SUCRL and it retains the advantage of temporal abstraction w.r.t. learning without options. Since FSUCRL does not require strong prior knowledge about options and its regret bound is partially computable, we believe the results of this paper could be used as a basis to construct more principled option discovery algorithms that explicitly optimize the exploration-exploitation performance of the learning algorithm.

## Acknowledgments

This research was supported in part by French Ministry of Higher Education and Research, Nord-Pas-de-Calais Regional Council and French National Research Agency (ANR) under project ExTra-Learn (n.ANR-14-CE24-0010-01).

## Footnotes

[1]Restricting the standard initial set to one state $s_o$ is without loss of generality (see App. A).

[2]Notice that $R_{\mathcal{O}}(s, o)$ (similarly for $\tau_{\mathcal{O}}$) is well defined only when $s = s_o$, that is when $o \in \mathcal{O}_s$.

[3]In the following we only focus on the dynamics of the process; similar definitions apply for the rewards.

[4]Notice that since option $o$ is defined in $s$, then $s = s_o$. Furthermore $\overline{r}$ is the MDP expected reward.

[5]Lem. 4 in App. D extends this result by giving an interpretation of $\mu_o(s')$, $\forall s' \in \mathcal{S}_o$.

[6]The provably smallest condition number (refer to [21, Th. 2.3]) is the one provided by Seneta [22]: $\widehat{\kappa}_{o,\min} = \tau_1(\widehat{Z}_o) = \max_{i,j}\frac{1}{2}\|\widehat{Z}_o(i,:) - \widehat{Z}_o(j,:)\|_1$ where $\widehat{Z}_o(i,:)$ is the $i$-th row of $\widehat{Z}_o = (I - \widehat{P}'_o + \mathbf{1}^{\mathsf{T}}\widehat{\mu}_o)^{-1}$.

[7]The confidence intervals on $\widetilde{P}'_o$ can never exclude a non-zero transition between any two states of $\mathcal{S}_o$. Therefore, the corresponding bounded-parameter MDP is always communicating and $\rho_o^*(u_j)$ is state-independent.

[8]We use $v_j$ instead of $u_j$ since the error in the inner EVI directly affects the value of the function at the outer EVI, which thus generates a sequence of functions different from $(u_j)$.

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
