[Supplementary Material]

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

[10]Note that the optimistic stationary distribution $\widetilde{\mu}_{s,o}$ is always uniquely defined since the optimistic transition matrix $\widetilde{P}_{s,o}$ outputted by the inner EVI algorithm is always unichain as is explained by Jaksch et al. [15]: there exists a state $s' \in \mathcal{S}_o$ such that the $s'$-th column of $\widetilde{P}_{s,o}$ has only strictly positive entries. This is always the case no matter whether we use Hoeffding bounds and the algorithm presented on Fig. 1 or Bernstein bounds and the algorithm presented on Fig. 2. Therefore, distribution $\widetilde{\mu}_{s,o}$ is the unique solution of $\widetilde{\mu}_{s,o}^\intercal \widetilde{P}_{s,o} = \widetilde{\mu}_{s,o}^\intercal$ and $\widetilde{\mu}_{s,o}^\intercal \mathbf{1} = 1$.

[11]To bound $\|\widetilde{\mu}_{s,o}^k-\mu_{s,o}\|_1$ we use the bound of Cho and Meyer [20] and introduce the condition number $\kappa_{s,o}^1$. This is possible because as already mentioned in footnote 10, the Markov Chain $\widetilde{P}_{s,o}$ is unichain and so the bound holds (it would not be the case if the Markov Chain had several recurrent classes).

[12] To estimate $b_\tau$, we can look at the terms $\mathbb{E}\left[(\tau - \overline{\tau})^k\right]$ for $k = 3, 4, ...$, and check that they are upper-bounded by $\frac{1}{2}k!\sigma_\tau^2 b_\tau^{k-2}$ (this corresponds to "Bernstein's condition").

[13] We chose the smallest condition numbers in the list of Cho and Meyer [20]. All the other condition numbers for the $\ell_1$-norm of $\mu$ are provably bigger or equal than the one we are using [21, Th. 2.3].

[14]The code used for the experiments is available on Github (https://github.com/RonanFR/UCRL).

[15]We need to distinguish between Hoeffding and Bernstein concentration inequalities. In the former case, we shrink directly the range, e.g., Eq. (6a) becomes $\left|r(s,a) - \widehat{r}_k(s,a)\right| \leq \alpha_r r_{\max} \sqrt{\frac{\log(SAt_k/\delta)}{N_k(s,a)}}$ . Bernstein bound is characterized by two terms: $\mathcal{O}(\sqrt{1/N_k} + 1/N_k)$. We have decided to shrink only the second term since the first one already scales with the empirical variance. For example, Eq. (6b) becomes $\left|p(s'|s,o) - \widehat{p}_k(s'|s,o)\right| \leq$

[16]$T_{\max}$ is the maximal *actual* duration as opposed to the *maximal* expected duration $\tau_{\max} \leq T_{\max}$.

[17]Here we have that $\Delta_R \approx 0$.

[18]We think that the settings of SUCRLv4 and SUCRLv5 are unrealistic since they assume to know exactly the reward function.

[19]For example assume you want to go out of the room through the first door and you start just in front of that door, the states located at the opposite side of the room are very unlikely to be reached by following the policy of the option, roughly $10^{-10}$ of the time. However, if you start that same option's policy from the opposite side of the room, then the states that were previously unlikely become very likely.

[20]We have empirically observed that the estimated condition numbers are close to the true one. This suggests that the problem is intrinsic in the definition of the bound in Eq. 8 and not due to bad estimates.

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

# A Assumptions

In this section, we discuss more in detail the assumptions used throughout the paper.

**Initial state** $s_o$**.** In its original definition [1], an option $o$ is characterized by an initial set $\mathcal{I}_o \subseteq \mathcal{S}$ where the policy $\pi_o$ is defined and can be executed. In Sect. 2 we restricted this set to $\mathcal{I}_o = \{s_o\}$. We can show that this restriction comes without loss of generality. Let $\mathcal{O}$ be a set of options with arbitrarily initial set $\mathcal{I}_o$, then we define an associated set of options $\mathcal{O}' = \{o' = \{\{s_{o'}^i\}, \beta_o, \pi_o\}, \forall o \in \mathcal{O}, \forall s_{o'}^i \in \mathcal{I}_o\}$. In other words, for all options $o \in \mathcal{O}$, we create $|\mathcal{I}_o|$ options with the same policy and termination condition, but with a starting state that corresponds to a single element of $\mathcal{I}_o$ and that we denote by $s_{o'}^i$ ($i$ for "initial" state).

$$\forall o \in \mathcal{O}, \ \mathcal{I}_{o'} = \{s_{o'}^i\}$$
$$\beta_{o'} = \beta_o$$
$$\pi_{o'} = \pi_o$$

It is immediate to notice that duplicating options does not change the behavior of a learning agent but only relabel the "name" of the options.

**Assumption 2 (initial state as terminal state).** Let $\mathcal{O}$ be a given set of options not satisfying the first part of Asm. 2 and $\mathcal{O}'$ the set of options obtained by forcing $\beta_o(s_o) = 1$ for all options in $\mathcal{O}$. It is straightforward to prove the following equivalence.

**Proposition 1.** *Let $\pi$ be a* stationary *(possibly randomized) policy over options $\mathcal{O}$. There exists a* stationary *policy $\pi'$ over options $\mathcal{O}'$ such that the process over states and actions is the same, i.e., for any sequence $\xi = (s_1, a_1, \dots,)$ then $\mathbb{P}_\pi(\xi) = \mathbb{P}_{\pi'}(\xi)$, where the transition probabilities are as in the primitive-action MDP.*

*Proof.* For any option $o \in \mathcal{O}$ in the orginal set of options, let's denote by $o' \in \mathcal{O}'$ the same option after forcing $\beta_o(s_o) = 1$ ($o'$ belongs to $\mathcal{O}'$). For any stationary policy $\pi$ over $\mathcal{O}$, let's define a corresponding stationary policy $\pi'$ over $\mathcal{O}'$ by: $\pi'(s) = (\pi(s))'$, $\forall s \in \mathcal{S}$. For any option $o$ such that $\pi(s_o) = o$ and $\beta_o(s_o) < 1$, the state $s_o$ might be visited while $o$ is being executed and $o$ is not stopped in $s_o$. But since $\pi_o$ (policy of the option $o$) is stationary Markov, the distribution on the sequence of states and actions visited after $s_o$ is exactly the same as if the option was first stopped and executed again (in both cases the policy $\pi_o$ and the starting state $s_o$ are the same). So the process over states and actions is the same for $\pi$ and $\pi'$. $\qquad\square$

This directly implies that the diameter over options is preserved (i.e., $D_\mathcal{O} = D_{\mathcal{O}'}$) as well as the optimal gain (i.e., $\rho_\mathcal{O}^* = \rho_{\mathcal{O}'}^*$). The only difference introduced in using $\mathcal{O}'$ is in case of non-stationary policies. FSUCRL generates a piece-wise stationary policy (composed by the policies generated over episodes) and an episode may indeed terminate before with $\mathcal{O}'$ than with $\mathcal{O}$. For instance, if an option $o$ is being executed and the episode termination condition of FSUCRL is met when arriving at $s_o$, with the set $\mathcal{O}$ the episode would continue, while with $\mathcal{O}'$ the condition $\beta_o(s_o) = 1$ would interrupt both the option and the episode. Since the number of episodes is small (i.e., logarithmic in the time horizon) and the difference in termination is limited, the overall impact of this assumption is negligible. As a result, whenever a set $\mathcal{O}$ does not satisfy this condition, we can simply run FSUCRL on the associated $\mathcal{O}'$.

**Assumption 2 (terminal states and inner states).** We first notice that this part of the assumption is similar in nature to the communicating assumption in UCRL. Let consider the case when the state space $\mathcal{S}$ contains a state $\bar{s}$ that is not reachable under any policy, then UCRL would have linear regret as $\bar{s}$ would be assigned a reward $r_{\max}$ which would never decrease, thus creating an optimistic policy trying to reach $\bar{s}$ without ever reaching it (see for example section 3 of [24]). Similarly, whenever $\beta_o(s) < 1$ implicitly "declares" that $s$ is reachable under $\pi_o$. If this is not the case, optimism at the level of option may attribute an average reward of $r_{\max}$ to the option, which would then be always executed under the optimistic policy, thus effectively making the problem *non-learnable*. Even when this assumption is not initially verified, we can easily change the terminal conditions to recover it. Given a set $\mathcal{O}$, we could run FSUCRL over a first phase of finite length. At the end of this phase $\mathcal{O}$ is turned into a new set of options $\mathcal{O}'$ where $\beta_o(s)$ is set to 1 for all states $s$ that have never been encountered while executing option $o$ during the initial phase. The resulting set $\mathcal{O}'$ would satisfy Asm. 2 by default. Furthermore, $\mathcal{O}'$ would always be "safer" than $\mathcal{O}$ since options would only be

shorter (we potentially increase the probability of termination over some states), thus potentially reducing the diameter $D'_\mathcal{O}$ and increasing the gain $\rho^*_{\mathcal{O}'}$, at the cost of reducing the temporal abstraction $T_n/n$.

## B Computing the Optimistic Transition Probabilities

In this section, we report the algorithms used by extended value iteration to compute the optimistic transition probabilities. All algorithms are based on the same intuitive idea: the higher the value of a state, the more probability mass should be allocated (while still satisfying the constraints imposed by the confidence bounds). While Alg. 1 is designed to handle Hoeffding bounds, Alg. 2 is specific to Bernstein confidence intervals.

If we consider Eq. 6b, we will use Alg. 2 for the computation of $\widetilde{\boldsymbol{b}}_o$ with parameters $\widehat{\boldsymbol{b}}_o = (\widehat{p}_k(s'|s,o))_{s'}$, $\beta^p_k$, and $\boldsymbol{u_j}$. As mentioned in the main paper, when we consider FSUCRLv1, the maximum over $\widehat{\boldsymbol{\mu}}_o$ in Eq. 7 can be solved by applying Alg. 1 with parameters $\widehat{\boldsymbol{\mu}}_o$, $\beta^\mu_k(o)$, and $\boldsymbol{\zeta}_o$.

---

**Algorithm 1** Optimistic Transition Probabilities (Hoeffding Bound) [15]

---

**Input:** Probability estimate $\hat{\boldsymbol{p}} \in \mathbb{R}^m$, confidence interval $d \in \mathbb{R}$, value vector $\boldsymbol{v} \in \mathbb{R}^m$
**Output:** Optimistic probabilities $\hat{\boldsymbol{p}}^+ \in \mathbb{R}^m$

Let $\mathcal{I} = \{i_1, i_2, \ldots, i_m\}$ such that $v_{i_1} \geq v_{i_2} \geq \ldots \geq v_{i_m}$
$p^+_{i_1} = \min\left\{1, \hat{p}_{i_1} + \frac{d}{2}\right\}$
$p^+_{i_j} = \hat{p}_{i_j}, \quad \forall j > 1$
$j = m$
**while** $\sum_{z=1}^m p^+_z > 1$ **do**
  $p^+_{i_j} = \max\left\{0, 1 - \sum_{z \neq i_j} p^+_z\right\}$
  $j = j - 1$
**end while**

---

**Algorithm 2** Optimistic Transition Probabilities (Bernstein Bound) [19]

---

**Input:** Probability estimate $\hat{\boldsymbol{p}} \in \mathbb{R}^m$, confidence intervals $\boldsymbol{d} \in \mathbb{R}^m$, value vector $\boldsymbol{v} \in \mathbb{R}^m$
**Output:** Optimistic probabilities $\hat{\boldsymbol{p}}^+ \in \mathbb{R}^m$

Let $\mathcal{I} = \{i_1, i_2, \ldots, i_m\}$ such that $v_{i_1} \geq v_{i_2} \geq \ldots \geq v_{i_m}$
$p^+_j = \max\{0, \hat{p}_j - d_j\}, \quad \forall j \in \{1, \ldots, m\}$
$\Delta = 1 - \sum_{j=1}^m p^+_j$
$j = 1$
**while** $\Delta > 0$ **do**
  $k = i_j$
  $\Delta' = \min\{\Delta, \hat{p}_k + d_k - p^+_k\}$
  $p^+_k = p^+_k + \Delta'$
  $\Delta = \Delta - \Delta'$
  $j = j + 1$
**end while**

---

## C Auxiliary Results and Proofs

### C.1 Proof of Lemma 1

The irreducible Markov chain of option $o$ has a finite number of states and is thus recurrent positive (see e.g., Thm. 3.3 in [25]). Moreover, $1/\mu_o(s)$ corresponds to the mean return time in a state $s$, i.e., the expected time to reach $s$ starting from $s$ (see e.g., Theorem 3.2 in [25]). Finally, $\overline{\tau}(s_o, o)$ is the expected time before reaching an absorbing states starting from $s_o$ in the original absorbing Markov chain $P_o$. Since in the irreducible Markov Chain all absorbing states are merged with $s_o$, $1/\mu_o(s_o)$ is

exactly equal to $\overline{\tau}(s_o, o)$ by definition. Let $(s_t)_{t\in\mathbb{N}}$ be the sequence of states visited while executing the irreducible Markov Chain starting from $s_0$ and let $\overline{r}_t = \overline{r}(s_t, \pi_o(s_t))$. By the Ergodic Theorem for Markov chains (see e.g., Thm. 4.1 in [25]):

$$\lim_{T\to+\infty} \frac{\sum_{t=0}^{T-1} \overline{r}_t}{T} = \sum_{s'\in\mathcal{S}_o} \overline{r}(s', \pi_o(s'))\mu_{s,o}(s') \quad \text{a.s.} \tag{15}$$

Let $T_0 = 0, T_1, T_2, ...$ be the successive times of visit to $s_o$ (random stopping times). From the Regenerative Cycle Theorem for Markov chains (see e.g., Thm. 7.4 in [26]) we have that the pieces of trajectory $(s_{T_n}, ..., s_{T_{n+1}-1})$, $n \geq 0$ are i.i.d. By the law of large numbers we thus have:

$$\frac{\sum_{t=0}^{T_n-1} \overline{r}_t}{n} = \frac{\sum_{k=0}^{n-1}\left(\sum_{t=T_k}^{T_{k+1}-1} \overline{r}_t\right)}{n} \xrightarrow[n\to+\infty]{} \overline{R}(s, o) \quad \text{a.s.}$$

Similarly, we have:

$$\frac{T_n}{n} = \frac{\sum_{k=0}^{n-1}(T_{k+1} - T_k)}{n} \xrightarrow[n\to+\infty]{} \overline{\tau}(s, o) \quad \text{a.s.}$$

By taking the ratio, the term $n$ disappears and we obtain:

$$\frac{\sum_{t=0}^{T_n-1} \overline{r}_t}{T_n} \xrightarrow[n\to+\infty]{} \frac{\overline{R}(s, o)}{\overline{\tau}(s, o)} \quad \text{a.s.} \tag{16}$$

All sub-sequences of a convergent sequence converge to the limit of that sequence. Extracting the subsequence $(T_n)_{n\in\mathbb{N}}$ in (15) we obtain:

$$\frac{\sum_{t=0}^{T_n-1} \overline{r}_t}{T_n} \xrightarrow[n\to+\infty]{} \sum_{s'\in\mathcal{S}_{s,o}} \overline{r}(s', \pi_o(s'))\mu_{s,o}(s') \quad \text{a.s.} \tag{17}$$

By uniqueness of the limit ((16) and (17)): $\overline{R}(s, o)/\overline{\tau}(s, o) = \sum_{s'\in\mathcal{S}_{s,o}} \overline{r}(s', \pi_o(s'))\mu_{s,o}(s')$.

## C.2 Proof of Lemma 2

As mentioned in the introduction, any communicating MDP together with a set of admissible options is associated to a communicating SMDP. We recall different equivalent formulations of the optimality equation for the induced SMDP:

$$u^*(s) = \max_{o\in\mathcal{O}_s}\left\{\overline{R}(s, o) - \rho^*\overline{\tau}(s, o) + \sum_{s'\in\mathcal{S}_\mathcal{O}} p(s'|s, o)u^*(s')\right\}$$

$$\overset{(a)}{\Leftrightarrow} u^*(s) = \max_{o\in\mathcal{O}_s}\left\{\frac{\overline{R}(s, o)}{\overline{\tau}(s, o)} - \rho^* + \frac{1}{\overline{\tau}(s, o)}\sum_{s'\in\mathcal{S}_\mathcal{O}} p(s'|s, o)u^*(s') - \frac{u^*(s)}{\overline{\tau}(s, o)}\right\} + u^*(s)$$

$$\overset{(b)}{\Leftrightarrow} \rho^* = \max_{o\in\mathcal{O}_s}\left\{\frac{\overline{R}(s, o)}{\overline{\tau}(s, o)} + \frac{1}{\overline{\tau}(s, o)}\left(\sum_{s'\in\mathcal{S}} p(s'|s, o)u^*(s') - u^*(s)\right)\right\} \tag{18}$$

$$\overset{(c)}{\Leftrightarrow} \rho^* = \max_{o\in\mathcal{O}_s}\left\{\sum_{s'\in\mathcal{S}_o} \overline{r}_o(s')\mu_o(s') + \mu_o(s)\left(\boldsymbol{b_o}^\mathsf{T}\boldsymbol{u^*} - u^*(s)\right)\right\},$$

where $(a)$ is obtained by data transformation, $(b)$ is obtained by reordering, and $(c)$ follows from Lem. 1 where $\overline{r}_o(s') \overset{def}{=} \overline{r}(s', \pi_o(s'))$ and $\boldsymbol{b_o} \overset{def}{=} (p(s'|s, o))_{s'\in\mathcal{S}}$. The optimality equation in Eq. 5 is directly derived from the last formulation when the reward $r_o$, the SMDP transition probabilities $\boldsymbol{b_o}$, and the stationary distribution of the associated irreducible MC $\mu_o$ are replaced by parameters $\widetilde{r}_o$, $\widetilde{\boldsymbol{b_o}}$, and $\widetilde{\mu}_o$ in suitable confidence intervals.

*Proof.* We propose a data transformation where any value $\widetilde{r}_o \in \mathcal{R}$, $\widetilde{\boldsymbol{b_o}} \in \mathcal{P}$, and $\widetilde{\mu}_o \in \mathcal{M}$ is mapped into the reward and transition of an equivalent MDP defined as

$$\forall s \in \mathcal{S}, \forall o \in \mathcal{O}_s, \begin{cases} \widetilde{r}^{\text{eq}}(s, o) \longleftarrow \sum_{s'\in\mathcal{S}_{s,o}} \widetilde{r}_o(s')\widetilde{\mu}_{s,o}(s') \\ \widetilde{p}^{\text{eq}}(s'|s, o) \longleftarrow \widetilde{\mu}_{s,o}(s)(\widetilde{b}_{s,o}(s') - \delta_{s,s'}) + \delta_{s,s'}, \quad \forall s' \in \mathcal{S}. \end{cases} \tag{19}$$

Since $\widetilde{r}^{\mathrm{eq}}(s,o)$ and $\widetilde{p}^{\mathrm{eq}}(s'|s,o)$ are continuous functions of $\widetilde{\boldsymbol{\mu}}_{s,o}$, $\widetilde{\boldsymbol{b}}_{s,o}$ and $\widetilde{\boldsymbol{r}}_{o}$, the compact sets $\mathcal{R}$, $\mathcal{P}$, and $\mathcal{M}$ are mapped into compact sets $\mathcal{R}^{\mathrm{eq}}$ and $\mathcal{P}^{\mathrm{eq}}$. Notice that this is exactly the same type of transformation applied at step $(a)$ of Eq. 18. More precisely, we obtain any equivalent bounded-parameter MDP with compact action spaces and communicating. As a result, we can directly apply the result from [27] and obtain the existence of a solution pair $(\widetilde{\rho}^*, \widetilde{u}^*)$ and the uniqueness of the optimal gain $\widetilde{\rho}^*$. Finally, the optimistic statement trivially follows from the fact that $\mathcal{R}$, $\mathcal{P}$, and $\mathcal{M}$ contain the true parameters of the MDP with options and Eq. 5 is taking a maximum over a larger set. $\qquad\square$

### C.3 Basic Results for (Extended) Value Iteration

We recall some basic properties of value iteration for average reward optimization in MDPs.

**Proposition 2.** *Let $\mathcal{L}$ be the (average reward) value iteration operator for any MDP $M$ such that for any bias function $\boldsymbol{u}$ and any state $s \in \mathcal{S}$*

$$\mathcal{L}u(s) = \max_{a \in \mathcal{A}_s} \Big\{ r(s,a) + \sum_{s' \in \mathcal{S}} p(s'|s,a)u(s') \Big\}. \tag{20}$$

*Then $\mathcal{L}$ is a non-expansion in both $\ell_\infty$-norm and in the span semi-norm, i.e., for any $\boldsymbol{u}, \boldsymbol{v}$*

$$\big\| \mathcal{L}\boldsymbol{u} - \mathcal{L}\boldsymbol{v} \big\|_\infty \le \|\boldsymbol{u} - \boldsymbol{v}\|_\infty, \tag{21}$$

$$\big\| \mathcal{L}\boldsymbol{u} - \mathcal{L}\boldsymbol{v} \big\|_{\mathrm{sp}} \le \|\boldsymbol{u} - \boldsymbol{v}\|_{\mathrm{sp}}. \tag{22}$$

*Proof.* While these properties are proved in [27], we recall the proof of Eq. 22 for completeness. For any bias functions $\boldsymbol{u}, \boldsymbol{v}$, and any state $s \in \mathcal{S}$

$$\mathcal{L}v(s) - \mathcal{L}u(s) \le \max_{a \in A_s} \Big\{ \sum_{s' \in \mathcal{S}} p(s'|s,a)(v(s') - u(s')) \Big\} \le \max_{s' \in \mathcal{S}}(v(s') - u(s')),$$

where the first inequality holds from $\max_x f(x) - \max_x g(x) \le \max_x(f(x) - g(x))$ and the second by maximizing over $s'$ and using $\sum_{s'} p(s'|s,a) = 1$. Symmetrically we have

$$\mathcal{L}u(s) - \mathcal{L}v(s) \le \max_{s' \in \mathcal{S}}(u(s') - v(s')),$$

and thus

$$
\begin{aligned}
\big\| \mathcal{L}\boldsymbol{u} - \mathcal{L}\boldsymbol{v} \big\|_{\mathrm{sp}} &\le \max_s(\mathcal{L}u(s) - \mathcal{L}v(s)) - \min_s(\mathcal{L}u(s) - \mathcal{L}v(s)) \\
&= \max_s(\mathcal{L}u(s) - \mathcal{L}v(s)) + \max_s(\mathcal{L}v(s) - \mathcal{L}u(s)) \\
&\le \max_s(u(s) - v(s)) + \max_s(v(s) - u(s)) = \|\boldsymbol{u} - \boldsymbol{v}\|_{\mathrm{sp}}.
\end{aligned}
$$

$\qquad\square$

**Proposition 3.** *The (average reward) value iteration for any MDP $M$ starts from an arbitrary bias function $\boldsymbol{u_0}$ and at each iteration $j$ computes*

$$u_{j+1}(s) = \max_{a \in \mathcal{A}_s} \Big\{ r(s,a) + \sum_{s' \in \mathcal{S}} p(s'|s,a)u_j(s') \Big\} \quad \Leftrightarrow \quad \boldsymbol{u_{j+1}} = \mathcal{L}\boldsymbol{u_j}. \tag{23}$$

*If $M$ is communicating with optimal gain $\rho^*$, then for any $s \in \mathcal{S}$*

$$\lim_{j \to \infty} \big( u_{j+1}(s) - u_j(s) \big) = \rho^*.$$

*Furthermore,* [9]

$$|g(\boldsymbol{u_{j+1}} - \boldsymbol{u_j}) - \rho^*| \le \|\boldsymbol{u_{j+1}} - \boldsymbol{u_j}\|_{\mathrm{sp}}/2$$

$$and \ |g(\boldsymbol{u_{j+1}} - \boldsymbol{u_j}) - \rho^{d_j}| \le \|\boldsymbol{u_{j+1}} - \boldsymbol{u_j}\|_{\mathrm{sp}}/2$$

*where $d_j$ is a greedy policy associated to $\boldsymbol{v_j}$ i.e., $\mathcal{L}_{d_j}\boldsymbol{v_j} = \mathcal{L}\boldsymbol{v_j}$.*

The previous properties hold for bounded parameter MDPs as well when actions, rewards, and transition probabilities belong to compact sets. As a result, for any state-action pair $s, a$, let $\mathcal{R}_{s,a}$, $\mathcal{P}_{s,a}$ be the compact set that rewards and transition probabilities belong to. Then extended value iteration

$$u_{j+1}(s) = \max_{a \in A_s} \max_{\widetilde{r}(s,a) \in \mathcal{R}_{s,a}} \left\{ \widetilde{r}(s,a) + \max_{\widetilde{p} \in \mathcal{P}_{s,a}} \sum_{s' \in \mathcal{S}} \widetilde{p}(s'|s,a) u_j(s') \right\}, \tag{24}$$

converges to the optimal gain $\rho^*$ of the corresponding bounded parameter MDP.

### C.4 Convergence Guarantees of FSUCRLv2

**Theorem 2.** *For any sequence of errors $(\varepsilon_j)_{j \geq 0}$ such that $\sum_{j \geq 0} \varepsilon_j = \Lambda < +\infty$, the nested EVI algorithm converges in the sense that the sequence $(\boldsymbol{v_j} - j\widetilde{\rho}^* \boldsymbol{1})_{j \geq 0}$ has a limit for any intial vector $v_0$. Therefore, the stopping condition $\|\boldsymbol{v_{j+1}} - \boldsymbol{v_j}\|_{\mathrm{sp}} + \frac{3}{2}\varepsilon_j \leq \varepsilon$ is always met in finite time and if it is met at step $j$ then*

1. *For all $s \in \mathcal{S}$, $|v_{j+1}(s) - v_j(s) - g(\boldsymbol{v_{j+1}} - \boldsymbol{v_j})| \leq \varepsilon$,*

2. *If $d_j$ is the policy returned by nested EVI, for all $s \in \mathcal{S}$:*

$$\left| \mathcal{L}_{d_j} v_j(s) - v_j(s) - g(\boldsymbol{v_{j+1}} - \boldsymbol{v_j}) \right| \leq \varepsilon$$

3. *$|\widetilde{\rho}^* - g(\boldsymbol{v_{j+1}} - \boldsymbol{v_j})| \leq \varepsilon/2$*

4. *If $\boldsymbol{v_0} = \boldsymbol{0}$ : $\|\boldsymbol{v_j}\|_{\mathrm{sp}} \leq D' + \Lambda$.*

*Proof.* **Step 1 (Convergence of the inner extended value iteration algorithm).** In order to simplify the notation, we denote by $\widehat{\rho}_o^*(v_j)$ the solution returned by the inner EVI at the stopping condition (Eq. 11), i.e.,

$$\widehat{\rho}_o^*(v_j) = g\big(\boldsymbol{w}_{j,l_j^o+1}^{\boldsymbol{o}} - \boldsymbol{w}_{j,l_j^o}^{\boldsymbol{o}}\big).$$

Combining the convergence guarantees of Prop. 3 with the stopping condition of the inner EVI for each option $o$, we obtain

$$\left| \max_{o \in \mathcal{O}_s} \widehat{\rho}_o^*(v_j) - \max_{o \in \mathcal{O}_s} \widetilde{\rho}_o^*(v_j) \right| \leq \max_{o \in \mathcal{O}_s} \big\| \boldsymbol{w}_{j,k_j^{s,o}+1}^{\boldsymbol{s,o}} - \boldsymbol{w}_{j,k_j^{s,o}}^{\boldsymbol{s,o}} \big\|_{\mathrm{sp}} / 2 \leq \varepsilon_j/2. \tag{25}$$

**Step 2 (Convergence of the outer extended value iteration algorithm).** We first introduce the operator $\mathcal{L}$ used in Eq. 7, i.e., for any bias function $u$

$$\mathcal{L}u(s) = \max_{o \in \mathcal{O}_s} \left\{ \widetilde{\rho}_o^*(u) \right\} + u(s), \tag{26}$$

where $\widetilde{\rho}_o^*(u)$ is defined in Eq. 9. Similarly, the nested EVI can be seen as a sequence of applications of an approximate operator

$$\mathcal{L}_{j,w}v(s) = \max_{o \in \mathcal{O}_s} \widehat{\rho}_o^*(v) + v(s), \tag{27}$$

where $\widehat{\rho}_o^*(v)$ is obtained by iterating (12) with initial vectors $w = (w_{s,o})$ and stopping condition $\varepsilon_j$ (so $\mathcal{L}_{j,w}$ only depends on $w$ and $\epsilon_j$). As a result Eq. 25 directly implies that $\forall w, \forall j \geq 0$, $\mathcal{L}_{j,w}$ is an $\varepsilon_j/2$-approximations of $\mathcal{L}$, i.e., for any $v$

$$\|\mathcal{L}_{j,w}\boldsymbol{v} - \mathcal{L}\boldsymbol{v}\|_\infty \leq \frac{\varepsilon_j}{2} \tag{28}$$

We can then compare the two sequences $(\mathcal{L}^j \boldsymbol{v_0})_{j \geq 0}$ and $(\boldsymbol{v_j})_{j \geq 0}$ such that $\mathcal{L}^0 \boldsymbol{v_0} = \boldsymbol{v_0}$, $\mathcal{L}^{j+1}\boldsymbol{v_0} = \mathcal{L}\left(\mathcal{L}^j \boldsymbol{v_0}\right)$ is the exact EVI and $\boldsymbol{v_{j+1}} = \mathcal{L}_j \boldsymbol{v_j}$ is the approximated EVI with $(\mathcal{L}_j)_{j \geq 0} = (\mathcal{L}_{j,w_j})_{j \geq 0}$ for any arbitrary choice of sequence $(w_j)_{j \geq 0}$. We have the following series of inequalities

$$\forall j \geq 0, \ \|\boldsymbol{v_{j+1}} - \mathcal{L}^{j+1}\boldsymbol{v_0}\|_\infty = \|\mathcal{L}_j \boldsymbol{v_j} - \mathcal{L}\left(\mathcal{L}^j \boldsymbol{v_0}\right)\|_\infty = \|\mathcal{L}_j \boldsymbol{v_j} - \mathcal{L}\boldsymbol{v_j} + \mathcal{L}\boldsymbol{v_j} - \mathcal{L}\left(\mathcal{L}^j \boldsymbol{v_0}\right)\|_\infty$$

$$\leq \|\mathcal{L}_j \boldsymbol{v_j} - \mathcal{L}\boldsymbol{v_j}\|_\infty + \|\mathcal{L}\boldsymbol{v_j} - \mathcal{L}\left(\mathcal{L}^j \boldsymbol{v_0}\right)\|_\infty \quad \text{(Triangle inequality)}$$

$$\leq \frac{\varepsilon_j}{2} + \|\boldsymbol{v_j} - \mathcal{L}^j \boldsymbol{v_0}\|_\infty \quad \text{(using (28) and (21))}.$$

Unrolling the previous inequality down to $\|v_0 - \mathcal{L}^0 v_0\|_\infty = 0$ and using the boundedness of the cumulative errors we obtain

$$\|v_j - \mathcal{L}^j v_0\|_\infty \leq \frac{1}{2} \sum_{k=0}^{j-1} \varepsilon_k$$

and more generally for any $i \geq 0$,

$$\|v_j - \mathcal{L}^j v_i\|_\infty \leq \frac{1}{2} \sum_{k=i}^{j-1} \varepsilon_k \qquad (29)$$

Futhermore, from Theorem 9.4.5 of Puterman [27], we know that for any initial vector $v$:

$$\lim_{j \to +\infty} \mathcal{L}^{j+1} v - \mathcal{L}^j v = \widetilde{\rho}^* \mathbf{1} \qquad (30)$$

We will now prove that the same property holds for the sequence $v_{j+1} - v_j$. For any $i, j$ such that $i < j$ we have the following decomposition:

$$v_{j+1} - v_j - \widetilde{\rho}^* \mathbf{1} = \mathcal{L}^{j+1-i} v_i - \mathcal{L}^{j-i} v_i - \rho^* \mathbf{1} + v_{j+1} - \mathcal{L}^{j+1-i} v_i + \mathcal{L}^{j-i} v_i - v_j$$

Using the triangle inequality we obtain:

$$\|v_{j+1} - v_j - \widetilde{\rho}^* \mathbf{1}\|_\infty \leq \|\mathcal{L}^{j+1-i} v_i - \mathcal{L}^{j-i} v_i - \widetilde{\rho}^* \mathbf{1}\|_\infty + \|v_{j+1} - \mathcal{L}^{j+1-i} v_i\|_\infty \\ + \|\mathcal{L}^{j-i} v_i - v_j\|_\infty \qquad (31)$$

Let's first bound the last two terms appearing in 31 using 29:

$$\|v_{j+1} - \mathcal{L}^{j+1-i} v_i\|_\infty \leq \frac{1}{2} \sum_{k=i}^{j} \varepsilon_k \leq \frac{1}{2} \sum_{k=i}^{+\infty} \varepsilon_k \xrightarrow[i \to +\infty]{} 0$$

and similarly: $\|\mathcal{L}^{j-i} v_i - v_j\|_\infty \leq \frac{1}{2} \sum_{k=i}^{j-1} \varepsilon_k \leq \frac{1}{2} \sum_{k=i}^{+\infty} \varepsilon_k \xrightarrow[i \to +\infty]{} 0$

Let's take $\varepsilon > 0$. By definition of the limit, there exists an integer $I(\varepsilon) \geq 0$ such that for all $j > i \geq I(\varepsilon)$:

$$\|v_{j+1} - \mathcal{L}^{j+1-i} v_i\|_\infty \leq \frac{\varepsilon}{3}$$

$$\text{and } \|\mathcal{L}^{j-i} v_i - v_j\|_\infty \leq \frac{\varepsilon}{3}$$

Let's now bound the remaining term appearing in 31. For any (fixed) $i \geq 0$ we know from 30 that this term converges to 0 as $j \to +\infty$ i.e.,

$$\|\mathcal{L}^{j+1-i} v_i - \mathcal{L}^{j-i} v_i - \widetilde{\rho}^* \mathbf{1}\|_\infty \xrightarrow[j \to +\infty]{} 0$$

This implies that there exists $J(\varepsilon, i) > i$ such that for all $j \geq J(\varepsilon, i)$:

$$\|\mathcal{L}^{j+1-i} v_i - \mathcal{L}^{j-i} v_i - \widetilde{\rho}^* \mathbf{1}\|_\infty \leq \frac{\varepsilon}{3}$$

Let's take $i = I(\varepsilon)$ and define $N(\varepsilon) \overset{def}{=} J(\varepsilon, I(\varepsilon))$. For all $j \geq N(\varepsilon)$:

$$\|v_{j+1} - v_j - \widetilde{\rho}^* \mathbf{1}\|_\infty \leq \frac{\varepsilon}{3} + \frac{\varepsilon}{3} + \frac{\varepsilon}{3} = \varepsilon$$

In conclusion (since $\varepsilon$ was taken arbitrarily): for all $\varepsilon > 0$, there exists an integer $N(\varepsilon) \geq 0$ such that for all $j \geq N(\varepsilon)$, $\|v_{j+1} - v_j - \widetilde{\rho}^* \mathbf{1}\|_\infty \leq \varepsilon$. This is exactly the definition of convergence thus:

$$\lim_{j \to +\infty} v_{j+1} - v_j = \widetilde{\rho}^* \mathbf{1}$$

As a consequence, $\|v_{j+1} - v_j - \widetilde{\rho}^* \mathbf{1}\|_{\mathrm{sp}} \xrightarrow[j \to +\infty]{} 0$ and so the stopping condition is always met in finite time. We can prove that the sequence $(v_j - j\widetilde{\rho}^* \mathbf{1})_{j \geq 0}$ converges using similar arguments. We

first note that for any initial vector $\boldsymbol{v}$, $\left(\mathcal{L}^j \boldsymbol{v} - j\widetilde{\rho}^* \mathbf{1}\right)_{j \geq 0}$ has a limit as $j$ tends to infinity (see Theorem 9.4.4. of Puterman [27]) and therefore it is a Cauchy sequence i.e.,

$$\sup_{k \geq 0} \|\mathcal{L}^{j+k} \boldsymbol{v} - \mathcal{L}^j \boldsymbol{v} - k\widetilde{\rho}^* \mathbf{1}\|_\infty \xrightarrow[j \to +\infty]{} 0 \tag{32}$$

Using a similar decomposition as before we have for all $j > i$:

$$\sup_{k \geq 0} \|\boldsymbol{v_{j+k}} - \boldsymbol{v_j} - k\widetilde{\rho}^* \mathbf{1}\|_\infty \leq \sup_{k \geq 0} \|\mathcal{L}^{j+k-i}\boldsymbol{v_i} - \mathcal{L}^{j-i}\boldsymbol{v_i} - k\widetilde{\rho}^* \mathbf{1}\|_\infty + \sup_{k \geq 0} \|\boldsymbol{v_{j+k}} - \mathcal{L}^{j+k-i}\boldsymbol{v_i}\|_\infty$$
$$+ \|\mathcal{L}^{j-i}\boldsymbol{v_i} - \boldsymbol{v_j}\|_\infty$$

The last two terms can be bounded as before:

$$\sup_{k \geq 0} \|\boldsymbol{v_{j+k}} - \mathcal{L}^{j+k-i}\boldsymbol{v_i}\|_\infty \leq \sup_{k \geq 0} \frac{1}{2}\sum_{l=i}^{j+k} \varepsilon_l = \frac{1}{2}\sum_{l=i}^{+\infty} \varepsilon_l \xrightarrow[i \to +\infty]{} 0$$

and similarly: $\|\mathcal{L}^{j-i}\boldsymbol{v_i} - \boldsymbol{v_j}\|_\infty \leq \frac{1}{2}\sum_{l=i}^{j-1} \varepsilon_l \leq \frac{1}{2}\sum_{l=i}^{+\infty} \varepsilon_l \xrightarrow[i \to +\infty]{} 0$

Let's take $\varepsilon > 0$. By definition of the limit, there exists an integer $I(\varepsilon) \geq 0$ such that for all $j > i \geq I(\varepsilon)$:

$$\sup_{k \geq 0} \|\boldsymbol{v_{j+k}} - \mathcal{L}^{j+k-i}\boldsymbol{v_i}\|_\infty \leq \frac{\varepsilon}{3}$$

$$\text{and } \|\mathcal{L}^{j-i}\boldsymbol{v_i} - \boldsymbol{v_j}\|_\infty \leq \frac{\varepsilon}{3}$$

For any (fixed) $i \geq 0$ we know from 32 that:

$$\sup_{k \geq 0} \|\mathcal{L}^{j+k-i}\boldsymbol{v_i} - \mathcal{L}^{j-i}\boldsymbol{v_i} - k\widetilde{\rho}^* \mathbf{1}\|_\infty \xrightarrow[j \to +\infty]{} 0$$

This implies that there exists $J(\varepsilon, i) > i$ such that for all $j \geq J(\varepsilon, i)$:

$$\sup_{k \geq 0} \|\mathcal{L}^{j+k-i}\boldsymbol{v_i} - \mathcal{L}^{j-i}\boldsymbol{v_i} - k\widetilde{\rho}^* \mathbf{1}\|_\infty \leq \frac{\varepsilon}{3}$$

Let's take $i = I(\varepsilon)$ and define $N(\varepsilon) \stackrel{def}{=} J(\varepsilon, I(\varepsilon))$. For all $j \geq N(\varepsilon)$:

$$\|\boldsymbol{v_{j+k}} - \boldsymbol{v_j} - k\widetilde{\rho}^* \mathbf{1}\|_\infty \leq \frac{\varepsilon}{3} + \frac{\varepsilon}{3} + \frac{\varepsilon}{3} = \varepsilon$$

In conclusion, $\lim_{j \to +\infty} \sup_{k \geq 0} \|\boldsymbol{v_{j+k}} - \boldsymbol{v_j} - k\widetilde{\rho}^* \mathbf{1}\|_\infty = 0$ which means that $(\boldsymbol{v_j} - j\widetilde{\rho}^* \mathbf{1})_{j \geq 0}$ is a Cauchy sequence so it is convergent since $(\mathbb{R}, \|\cdot\|_\infty)$ is a complete metric space. Moreover, since $\varepsilon_j \xrightarrow[j \to +\infty]{} 0$ the limit must necessarily be a solution of the optimality equation.

**Step 3 (Validity of the stopping condition).** To prove the validity of the stopping condition, we adapt the proof from [27]. We start by the following Lemma based on Theorem 8.5.5 of [27]:

**Lemma 3.** *For any vector $\boldsymbol{v}$ and any decision rule $d_j$ achieving the maximum $\mathcal{L}_j \boldsymbol{v}$ we have:*

$$\forall s \in \mathcal{S}, \ \min_{s' \in \mathcal{S}} \{\mathcal{L}_j v(s') - v(s')\} - \frac{\varepsilon_j}{2} \leq \widetilde{\rho}^{d_j}(s) \leq \widetilde{\rho}^* \leq \max_{s' \in \mathcal{S}} \{\mathcal{L}_j v(s') - v(s')\} + \frac{\varepsilon_j}{2} \tag{33}$$

*Proof.* To prove this lemma, we use the same arguments as Puterman [27]:

$$\widetilde{\rho}^{d_j} = \widetilde{P}_{d_j}^* \widetilde{\boldsymbol{r}}_{d_j} = \widetilde{P}_{d_j}^* (\widetilde{\boldsymbol{r}}_{d_j} + \widetilde{P}_{d_j} \boldsymbol{v} - \boldsymbol{v}) \geq \widetilde{P}_{d_j}^* (\mathcal{L}_j \boldsymbol{v} - \boldsymbol{v}) - \frac{\varepsilon_j}{2} \mathbf{1}$$

$$\Rightarrow \forall s \in \mathcal{S}, \ \widetilde{\rho}^{d_j}(s) \geq \min_{s' \in \mathcal{S}} \{\mathcal{L}_j v(s') - v(s')\} - \frac{\varepsilon_j}{2}$$

where $\widetilde{P}_{d_j}^*$ is the limiting matrix of $\widetilde{P}_{d_j}$. The first inequality follows from Prop. 3 and the fact that $\widetilde{P}_{d_j}^* \mathbf{1} = \mathbf{1}$. Note that $d_j$ corresponds to the choice of both a policy of options and a value for the

parameters of the MDP (compact spaces), but this doesn't impact the proof. We know from Lemma 2 that there exists an optimal decision rule $\delta$ that achieves $\widetilde{\rho}^{\delta} = \widetilde{\rho}^* \mathbf{1}$ and so similarly:

$$\widetilde{\rho}^* \mathbf{1} = \widetilde{\rho}^{\delta} = \widetilde{P}_{\delta}^* \widetilde{r}_{\delta} = \widetilde{P}_{\delta}^* (\widetilde{r}_{\delta} + \widetilde{P}_{\delta} v - v) \leq \widetilde{P}_{\delta}^* (\mathcal{L} v - v) \leq \widetilde{P}_{\delta}^* (\mathcal{L}_j v - v) + \frac{\varepsilon_j}{2} \mathbf{1}$$

$$\Rightarrow \widetilde{\rho}^* \leq \max_{s' \in \mathcal{S}} \{\mathcal{L}_j v(s') - v(s')\} + \frac{\varepsilon_j}{2}$$

where the first inequality comes from the definition of $\mathcal{L}$ and the second inequality follows from (28). $\qquad\square$

If we apply Lemma 3 to $v_j$ then we have:

$$\|v_{j+1} - v_j\|_{\mathrm{sp}} + \varepsilon_j \geq \widetilde{\rho}^* - \max_{s \in \mathcal{S}} \{\rho^{d_j}(s)\}$$

Moreover, by definition of function $g$ we have:

$$\min_{s' \in \mathcal{S}} \{\mathcal{L}_j v_j(s') - v_j(s')\} - \frac{\varepsilon_j}{2} \leq g(v_{j+1} - v_j) \leq \max_{s' \in \mathcal{S}} \{\mathcal{L}_j v_j(s') - v_j(s')\} + \frac{\varepsilon_j}{2}$$

For any scalars $x$, $y$, and $z$, if $x \leq y \leq z$ and $z - x \leq \epsilon$:

$$-\frac{\epsilon}{2} \leq \frac{1}{2}(x - z) \leq y - \frac{1}{2}(x + z) \leq \frac{1}{2}(z - x) \leq \frac{\epsilon}{2}$$

Therefore by taking $x = \max_{s' \in \mathcal{S}} \{\mathcal{L}_j v_j(s') - v_j(s')\} + \frac{\varepsilon_j}{2}$, $z = \min_{s' \in \mathcal{S}} \{\mathcal{L}_j v_j(s') - v_j(s')\} - \frac{\varepsilon_j}{2}$ and $y = \widetilde{\rho}^*$ or $y = \widetilde{\rho}^{d_j}$ we obtain:

$$|\widetilde{\rho}^* - g(v_{j+1} - v_j)| \leq \frac{1}{2}(\|v_{j+1} - v_j\|_{\mathrm{sp}} + \varepsilon_j)$$

$$\text{and } |\widetilde{\rho}^{d_j} - g(v_{j+1} - v_j)| \leq \frac{1}{2}(\|v_{j+1} - v_j\|_{\mathrm{sp}} + \varepsilon_j)$$

When the stopping condition $\|v_{j+1} - v_j\|_{\mathrm{sp}} + \frac{3}{2}\varepsilon_j \leq \varepsilon$ holds, we have:

$$|\widetilde{\rho}^* - g(v_{j+1} - v_j)| \leq \frac{\varepsilon}{2} - \frac{\varepsilon_j}{2} \leq \frac{\varepsilon}{2}$$

Using the same argument as Lemma 7 of Fruit and Lazaric [14] we also have:

$$\forall s \in \mathcal{S}, \ |v_{j+1}(s) - v_j(s) - g(v_{j+1} - v_j)| \leq \varepsilon - \frac{\varepsilon_j}{2} \leq \varepsilon$$

Finally, by Prop. 3, we know that $\forall s \in \mathcal{S}, \ |\mathcal{L}_{d_j} v_j(s) - v_{j+1}(s)| \leq \varepsilon_j / 2$ implying:

$$\forall s \in \mathcal{S}, \ |\mathcal{L}_{d_j} v_j(s) - v_j(s) - g(v_{j+1} - v_j)| \leq |v_{j+1}(s) - v_j(s) - g(v_{j+1} - v_j)| + \frac{\varepsilon_j}{2}$$

$$\leq \varepsilon - \frac{\varepsilon_j}{2} + \frac{\varepsilon_j}{2} = \varepsilon$$

**Step 4 (Bound on the bias span).** Using the same argument as in [15] and [14] we can show that when $v_0 = 0$, $\|\mathcal{L}^j v_0\|_{\mathrm{sp}} \leq D'$. However, this property does not apply to $\|v_j\|_{\mathrm{sp}}$ since at every time step of value iteration, we potentially make a small error (either positive or negative) and so $v_j$ is no longer the maximal expected cumulative rewards after $j$ steps. Nevertheless, using the reverse triangle inequality, the fact that $\|\mathcal{L}^j v_0\|_{\mathrm{sp}} \leq D'$ and the inequality $\| \cdot \|_{\mathrm{sp}} \leq 2\| \cdot \|_{\infty}$ we have:

$$\forall j \geq 0, \ \|v_j\|_{\mathrm{sp}} - \|\mathcal{L}^j v_0\|_{\mathrm{sp}} \leq \|v_j - \mathcal{L}^j v_0\|_{\mathrm{sp}} \leq 2\|v_j - \mathcal{L}^j v_0\|_{\infty} \leq \Lambda$$

$$\Rightarrow \|v_j\|_{\mathrm{sp}} \leq \|\mathcal{L}^j v_0\|_{\mathrm{sp}} + \Lambda \leq D' + \Lambda$$

We already proved that $v_j - j\widetilde{\rho}^* \mathbf{1}$ is converging to a solution $v^*$ of the optimality equation $\mathcal{L} v^* = v^* + \widetilde{\rho}^* \mathbf{1}$ and as a consequence of Theorem 4 of Bartlett and Tewari [28], such a solution $v^*$ satisfies $\|v^*\|_{\mathrm{sp}} \leq D'$. This mean that

$$\|v_j\|_{\mathrm{sp}} = \|v_j - j\widetilde{\rho}^* \mathbf{1}\|_{\mathrm{sp}} \xrightarrow[j \to +\infty]{} \|v^*\|_{\mathrm{sp}} \leq D'$$

and thus as $j$ grows, the bound $\|v_j\|_{\mathrm{sp}} \leq D' + \Lambda$ will eventually become loose (more specifically the term $\Lambda$ can be dropped). The term $\|\mathcal{L} v_j - v_j\|_{\mathrm{sp}}$ is the quantity used to characterize the error in gain

Figure 4: Counter-example showing that the inequality $\|\boldsymbol{v_j}\|_{\mathrm{sp}} \leq \|\mathcal{L}\boldsymbol{v_j} - \boldsymbol{v_j}\|_{\mathrm{sp}} + \|\boldsymbol{v}^*\|_{\mathrm{sp}}$ is not true in general (where $\boldsymbol{v_j}$ is any vector, $\mathcal{L}$ is the optimal Bellman operator and $\boldsymbol{v}^*$ is an optimal bias).

and one might wonder whether it could also be used to quantify the error in bias span $\|\boldsymbol{v_j}\|_{\mathrm{sp}} - \|\boldsymbol{v}^*\|_{\mathrm{sp}}$ i.e., $\|\boldsymbol{v_j}\|_{\mathrm{sp}} \leq \|\mathcal{L}\boldsymbol{v_j} - \boldsymbol{v_j}\|_{\mathrm{sp}} + \|\boldsymbol{v}^*\|_{\mathrm{sp}}$. The counter-example of Fig. 4 shows that this is not the case. Taking $\boldsymbol{v_0} = [2/\delta, 0]^\mathsf{T}$ we have:

$$\begin{cases} \boldsymbol{v}^* = [0, -1/\delta]^\mathsf{T} \\ \boldsymbol{v_1} = \mathcal{L}\boldsymbol{v_0} = [1 + 2/\delta, 2]^\mathsf{T} \end{cases} \implies \|\boldsymbol{v_1} - \boldsymbol{v_0}\|_{\mathrm{sp}} = 1 < \|\boldsymbol{v_0}\|_{\mathrm{sp}} - \|\boldsymbol{v}^*\|_{\mathrm{sp}} = 2/\delta - 1/\delta = 1/\delta$$

for $1 > \delta > 0$. It is thus impossible to quantify the error in bias span using the usual stopping condition based on $\|\mathcal{L}\boldsymbol{v_j} - \boldsymbol{v_j}\|_{\mathrm{sp}}$. Whether another stopping condition could be used is left as an open question.

$\square$

## D    Regret Proof

### D.1    Notations

To simplify notations in this proof, we will denote by $\square$ any numerical constant (which may vary from line to line). In all this section, we use the notations $M = \{\mathcal{S}, \mathcal{A}, r, p\}$ for the original MDP and $M' = \{\mathcal{S}', \mathcal{O}, R, b\}$ for the SMDP induced by the set of options $\mathcal{O}$. We also denote by $\mathcal{M}_k$ the set of MDPs with options compatible with the confidence intervals of (6). When this set contains the true MDP with options, we use the notations $M, M' \in \mathcal{M}_k$. To avoid ambiguity, we denote an option by the pair $(s, o)$ where $s$ is the starting state.

### D.2    Preliminary assumptions

To improve the readability of the proof, we will first make three simplifying assumptions and we later show why relaxing these assumptions has only a minor impact on the regret (see section D.8).

**Assumption 3.** *For the rest of the proof, we assume that:*

1. *The exact version of EVI (presented in equation (7)) is run i.e., $\forall j \geq 0, \varepsilon_j = 0$*

2. *In every episode, the first time step for which the number of visits of a state-action pair has doubled always occurs at the end of the execution of an option*

3. *All the irreducible Markov Chains corresponding to the options are aperiodic (hence ergodic)*

### D.3    Splitting into episodes

We denote by $k \in \{1...m\}$ the indices of the episodes of the algorithm, and by $t \in \{1...T\}$ the indices of the time steps (when a primitive action is executed). In contrast, the indices of the decision steps (when an option is executed) are denoted by $i \in \{1...n\}$. An episode $k$ starts at time $t = t_k$ and at decision step $i = i_k$. The random variables $s_t$ and $o_t$ denotes respectively the state visited at time $t$ and the option started or being executed at time $t$. By definition, the primitive action executed at time

$t$ is always $a_t = \pi_{o_t}(s_t)$. We can split the regret in two different terms:

$$\Delta = \left(\sum_{s,o}\sum_{k=1}^{m} t_k(s,o)\right)\rho^* - \sum_{i=1}^{n} R_i(s,o) = \sum_{k=1}^{m}\sum_{s,o}\left(t_k(s,o)\rho^* - \nu_k(s,o)\overline{\tau}(s,o)\overline{\omega}(s,o)\right)$$
$$+ \sum_{k=1}^{m}\sum_{s,o}\nu_k(s,o)\overline{\tau}(s,o)\overline{\omega}(s,o) - \sum_{i=1}^{n} R_i(s,o) \tag{34}$$

where $t_k(s,o) = \sum_{t=t_k}^{t_{k+1}-1}\mathbb{1}_{\{(s,o)_t=(s,o)\}}$ is the total amount of time steps spent executing option $o \in \mathcal{O}_s$ started in $s \in \mathcal{S}'$ during episode $k$, $R_i(s_i, o_i)$ is the total reward earned while executing option $o_i \in \mathcal{O}_s$ started in $s_i \in \mathcal{S}'$ at decision step $i$, and $\overline{\omega}(s,o) = \overline{R}(s,o)/\overline{\tau}(s,o)$. The time spent in $(s,o)$ before episode $k$ is: $T_k(s,o) = \sum_{j=1}^{k-1} t_j(s,o)$. Similarly to Jaksch et al. [15], we denote by $\nu_k(s,o)$ the total number of visits in state-option pair $(s,o) \in \mathcal{S}' \times \mathcal{O}_s$ (of the SMDP) during episode $k$, and $\nu_k(s,a)$ the total number of visits in state-action pair $(s,a) \in \mathcal{S} \times \mathcal{A}_s$ (of the original MDP). We define: $N(s,o) = \sum_{k=1}^{m}\nu_k(s,o)$ and $N(s,a) = \sum_{k=1}^{m}\nu_k(s,a)$. The analysis for the last term of (34) is the same as for SUCRL [14]. There exists an event $\Omega_0$ of probability greater than $1 - \delta$ for which:

$$\forall n \geq 1, \quad \sum_{s,o}\sum_{k=1}^{m}\left(\nu_k(s,o)\overline{\tau}(s,o)\overline{\omega}(s,o) - R_k(s,o)\right) = \sum_{s,o}N(s,o)\overline{R}(s,o) - \sum_{i=1}^{n} R_i(s,o)$$

$$\leq \begin{cases} \square\sigma_R\sqrt{n\log\left(\frac{\square n}{\delta}\right)} & \text{if } n \geq \square\frac{b_R^2}{\sigma_R^2}\log\left(\frac{\square n}{\delta}\right) \\ \square b_R\log\left(\frac{\square n}{\delta}\right) & \text{otherwise} \end{cases} \tag{35}$$

$$\leq \square\left(b_R\log\left(\frac{b_R\log\left(\frac{n}{\delta}\right)}{\delta\sigma_R}\right) + \sigma_R\sqrt{n\log\left(\frac{n}{\delta}\right)}\right)$$

Let's now analyse the remaining term in (34) decomposed over different episodes as

$$\Delta_k = \sum_{s,o}\left(t_k(s,o)\rho^* - \nu_k(s,o)\overline{\tau}(s,o)\overline{\omega}(s,o)\right)$$

### D.4   Dealing with failing confidence bounds

We assumed that the stopping condition of an episode is always met once all options are ended. Therefore, the stopping condition of an episode is strictly equivalent to the stopping condition used in UCRL2 [15] so there exists an event $\Omega_1$ of probability at least $1 - \delta$ for which (see Jaksch et al. [15] for the derivation):

$$\forall T \geq 1, \quad \sum_{k=1}^{m}\Delta_k\mathbb{1}_{\{M\notin\mathcal{M}_k\}} \leq \rho^*\sum_{s,o}\sum_{k=1}^{m} t_k(s,o)\mathbb{1}_{\{M\notin\mathcal{M}_k\}} = \rho^*\sum_{s,a}\sum_{k=1}^{m}\nu_k(s,a)\mathbb{1}_{\{M\notin\mathcal{M}_k\}}$$

$$\leq r_{\max}\sqrt{T}$$

### D.5   Dealing with mixing times of options

We study the impact on the regret of the speed of convergence of an option (seen as an irreducible Markov Chain using Lem. 1) to its stationary distribution. This corresponds to what we could call the "mixing time" of the option (by analogy to the MCMC literature). Let's first recall the Bernstein inequality for aperiodic Markov Chains:

**Theorem 3** (Thm. 3.4 and Prop. 3.10 of Paulin [29]). *Let $X_1, ..., X_n$ be a time-homogeneous, irreducible and aperiodic Markov Chain with initial probability distribution $\mu_0$, stationary distribution $\mu$ and pseudo-spectral gap $\gamma$. Let $f$ be a square-integral function over $\mu$ such that $|f(x) - \mathbb{E}_\mu[f]| \leq C$ for every $x$. Let $V_f = \mathrm{Var}_\mu(f)$ and $S_n = \sum_{i=1}^{n} f(X_i)$. We have:*

$$\forall\epsilon \geq 0, \ \mathbb{P}_{\mu_0}\left(|S_n - \mathbb{E}_\mu[S_n]| \geq \epsilon\right) \leq 2\sqrt{N_{\mu_0}}\exp\left(\frac{-\epsilon^2\gamma}{16(n+1/\gamma)V_f + 40C\epsilon}\right)$$

$$where \ N_{\mu_0} = \mathbb{E}_\mu\left[\left(\frac{d\mu_0}{d\mu}\right)^2\right] = \mathbb{E}_{\mu_0}\left[\frac{d\mu_0}{d\mu}\right]$$

The pseudo-spectral gap of a Markov Chain is of the order of the inverse of the mixing time [29]. For all $s \in \mathcal{S}'$, $o \in \mathcal{O}_s$, $s' \in \mathcal{S}_o$ and $k$, denote by $N_o^k(s') = \sum_{j=1}^{k-1} \sum_{t=t_k}^{t_{k+1}-1} \mathbb{1}_{\{(s,o)_t = (s,o),\ s_t = s'\}}$ the total number of visits in state $s'$ while executing option $(s,o)$ before episode $k$. Assume we ignore all time steps $t$ that do not satisfy $(s,o)_t = (s,o)$ i.e., we focus on the sequence of states when option $(s,o)$ is executed. This sequence of states is itself a Markov Chain. More precisely, $N_o^k(s')$ has the form $\sum_{i=1}^n f(X_i)$ where $(X_i)_i$ is the sequence of visited states in the ergodic Markov Chain representing the option and $f(X_i) = \mathbb{1}_{\{X_i = s'\}}$ while $T_k(s,o)\mu_{s,o}(s')$ corresponds to $n\mathbb{E}_\mu[f(X)]$. We can thus apply Theorem 3 where: $C = 1$, $V_f = \mu_{s,o}(s')(1 - \mu_{s,o}(s'))$, $n = T_k(s,o)$, $N_{\mu_0} = 1/\mu_{s,o}(s) = \overline{\tau}(s,o)$ ($\mu_0$ is a Dirac in the initial state of the option $s$). With probability at least $1 - \delta$:

$$T_k(s,o)\mu_{s,o}(s') - N_o^k(s') \le \square \left( \frac{\log(\overline{\tau}(s,o)/\delta)}{\gamma_{s,o}} + \sqrt{\frac{\mu_{s,o}(s')(1 - \mu_{s,o}(s'))}{\gamma_{s,o}} T_k(s,o) \log\left(\frac{\overline{\tau}(s,o)}{\delta}\right)} \right)$$

By adjusting $\delta$ and taking a union bound over all $s$, $o$, $s'$, $k$ and $T_k(s,o)$ we can create an event $\Omega_2$ of probability at least $1 - \delta$ for which:

$$\forall s, o, s' \forall k,\ T_k(s,o)\mu_{s,o}(s') - N_o^k(s') \le \square \left( \frac{1}{\gamma_{s,o}} \log\left(\frac{\overline{\tau}(s,o)kS_o S' OT_k(s,o)}{\delta}\right) \right.$$
$$\left. + \sqrt{\frac{\mu_{s,o}(s')(1 - \mu_{s,o}(s'))}{\gamma_{s,o}} T_k(s,o) \log\left(\frac{\overline{\tau}(s,o)kS_o S' OT_k(s,o)}{\delta}\right)} \right)$$

Note that $S_o \le S'$ so the term $S_o S'$ in the log can be replaced by $S'$ (the square becomes a multiplicative constant in front of the log). Let $\mu_{s,o}^* = \min_{s' \in \mathcal{S}_o}\{\mu_{s,o}(s')\}$ and define for all $s \in \mathcal{S}'$ and $o \in \mathcal{O}_s$:

$$T_{s,o} = \min \left\{ t \ge 1 : \square \left( \frac{\log(\tau_{\max} m S' OT/\delta)}{\mu_{s,o}^* \gamma_{s,o} t} + \sqrt{\frac{\log(\tau_{\max} m S' OT/\delta)}{\mu_{s,o}^* \gamma_{s,o} t}} \right) \le \frac{1}{2} \right\}$$
$$\le \square \left( \frac{\log(\tau_{\max} m S' OT/\delta)}{\mu_{s,o}^* \gamma_{s,o}} \right)$$

$$K_{s,o} = \{k \in \{1...m\} : T_k(s,o) < T_{s,o}\} \text{ and } m_{s,o} = \max\{K_{s,o}\}$$

Under event $\Omega_2$, if $T_k(s,o) \ge T_{s,o}$ (i.e., $k \notin K_{s,o}$), by definition of $T_{s,o}$:

$$\forall s' \in \mathcal{S}_o,\ \frac{T_k(s,o)\mu_{s,o}(s') - N_o^k(s')}{T_k(s,o)\mu_{s,o}(s')} \le \frac{1}{2}$$

$$\Rightarrow \forall s' \in \mathcal{S}_o,\ \frac{1}{N_o^k(s')} = \frac{1}{T_k(s,o)\mu_{s,o}(s')} \times \frac{1}{1 - \frac{T_k(s,o)\mu_{s,o}(s') - N_o^k(s')}{T_k(s,o)\mu_{s,o}(s')}} \le \frac{2}{T_k(s,o)\mu_{s,o}(s')}$$

where we used the fact that $\forall x \le 1/2$, $1/(1-x) \le 2$. In the rest of the proof we will use the above inequality to replace $1/N_o^k(s')$ by $1/T_k(s,o)\mu_{s,o}(s')$ in all episodes where $k \notin \bigcup_{s,o} K_{s,o}$.

Let $k(t)$ be the index of the episode at time step $t$. The regret resulting from all time steps where $k(t) \in K_{s_t, o_t}$ is:

$$r_{\max} \sum_{t=1}^T \mathbb{1}_{\{k(t) \in K_{s_t, o_t}\}} = r_{\max} \sum_{s,o} \sum_{k=1}^m \sum_{t=t_k}^{t_{k+1}-1} \mathbb{1}_{\{T_k(s,o) < T_{s,o}\}} \mathbb{1}_{\{s_t = s, o_t = o\}}$$

$$= r_{\max} \sum_{s,o} \sum_{k=1}^m \mathbb{1}_{\{T_k(s,o) < T_{s,o}\}} t_k(s,o)$$

$$= r_{\max} \sum_{s,o} \sum_{k=1}^m \mathbb{1}_{\{T_k(s,o) < T_{s,o}\}} (T_{k+1}(s,o) - T_k(s,o))$$

$$= r_{\max} \sum_{s,o} \sum_{k=1}^{m_{s,o}} (T_{k+1}(s,o) - T_k(s,o))$$

$$= r_{\max} \sum_{s,o} T_{m_{s,o}+1}(s,o) \le 2r_{\max} \sum_{s,o} T_{m_{s,o}}(s,o) \le 2r_{\max} \sum_{s,o} T_{s,o}$$

where the first inequality comes from the fact that $\forall k \geq 0,\ T_{k+1}(s,o) \leq 2T_k(s,o)$ due to the stopping condition of an episode, and the second inequality comes from the definition of $m_{s,o}$. In conclusion:

$$r_{\max} \sum_{t=1}^{T} \mathbb{1}_{\{k(t) \in K_{s_t,o_t}\}} \leq \square \left( \frac{r_{\max} SO}{\mu^\star \gamma^\star} \log \left( \frac{\tau_{\max} SAOT \log (T/SA)}{\delta} \right) \right)$$

where $\mu^\star = \min_{s,o}\{\mu^\star_{s,o}\}$ and $\gamma^\star = \min_{s,o}\{\gamma_{s,o}\}$ and we used $m \leq \square SA \log \left(\frac{T}{SA}\right)$ (as is proved by Jaksch et al. [15] where they use the same stopping condition of an episode).

## D.6 Episodes where $M, M' \in \mathcal{M}_k$ and $k \notin \bigcup_{s,o} K_{s,o}$

We define two optimistic average gains for an option: $\omega_k^+(s,o) = \sum_{s' \in \mathcal{S}_o} \widetilde{r}_k(s', \pi_o(s')) \mu_{s,o}(s')$ (true stationary distribution but optimistic rewards) and $\widetilde{\omega}_k(s,o) = \sum_{s' \in \mathcal{S}_o} \widetilde{r}_k(s', \pi_o(s')) \widetilde{\mu}_{s,o}(s')$ (optimistic stationary distribution[10] and optimistic rewards). We consider the following decomposition:

$$\Delta_k = \sum_{s,o} t_k(s,o) \left(\rho^* - \widetilde{\omega}_k(s,o)\right) + \sum_{s,o} t_k(s,o) \left(\widetilde{\omega}_k(s,o) - \omega_k^+(s,o)\right)$$
$$+ \sum_{s,o} \omega_k^+(s,o) \left(t_k(s,o) - \nu_k(s,o)\overline{\tau}(s,o)\right) + \sum_{s,o} \nu_k(s,o)\overline{\tau}(s,o) \left(\omega_k^+(s,o) - \overline{\omega}(s,o)\right) \tag{36}$$

**Lemma 4.** $\forall s \in \mathcal{S}',\ \forall o \in \mathcal{O}_s,\ \forall s' \in \mathcal{S}_o$, the quantity $\overline{\tau}(s,o)\mu_{s,o}(s')$ corresponds to the expected number of visits in $s'$ when $(s,o)$ is executed until termination.

*Proof.* This lemma extends Lem. 1. By Thm. 2.1. of [25], the following measure $\lambda_{s,o}$ is invariant for the irreducible Markov Chain of option $(s,o)$ (characterized by transition matrix $P'_{s,o}$):

$$\forall s' \in \mathcal{S}_o,\ \lambda_{s,o}(s') = \mathbb{E}\left[ \sum_{k=1}^{H(s)} \mathbb{1}_{\{s_k=s'\}} \middle| s_0 = s \right] \quad \text{where } H(s) = \inf\{k \geq 1 :\ s_k = s\}$$

$H(s)$ is the first return time in $s$. $\lambda_{s,o}$ is one of the regenerative forms of the invariant measures of $P'_{s,o}$. By definition, $\lambda_{s,o}(s')$ corresponds to the expected number of visits in $s'$ when starting in $s$ and before returning in $s$ i.e., in our case it is exactly the expected number of visits in $s'$ when $(s,o)$ is executed until termination. By Thm. 2.2. of Bremaud [25] $\lambda_{s,o}$ is proportional to $\mu_{s,o}$: $\lambda_{s,o} = C\mu_{s,o}$. By definition of $H(s)$ and using Lem. 1: $\lambda_{s,o}(s) = 1 = C\mu_{s,o}(s) = C/\overline{\tau}(s,o) \Rightarrow C = \overline{\tau}(s,o)$ which concludes the proof. $\square$

We note that $\overline{\omega}(s,o)$ and $\omega_k^+(s,o)$ are both discrete integrals over the true stationary distribution. Using Lemma 4, the last term of (36) can thus be expressed as a conditional expectation knowing the number of execution of every option at every episode:

$$\sum_{s,o} \sum_{k=1}^{m} \nu_k(s,o)\overline{\tau}(s,o) \left(\omega_k^+(s,o) - \overline{\omega}(s,o)\right) = \mathbb{E}\left[ \sum_{s,a} \sum_{k=1}^{m} \nu_k(s,a) \left(\widetilde{r}_k(s,a) - \overline{r}(s,a)\right) \middle| (\nu_k(s,o))_{k,s,o} \right]$$

Moreover, we compute the optimistic rewards $\widetilde{r}_k(s,a)$ in the same way as Jaksch et al. [15] (the confidence bounds used for the rewards and transition probabilities of the original MDP are the same) and so we know that there exists an event $\Omega_3$ of probability at least $1 - \delta$ such that for all values of $(\nu_k(s,o))_{k,s,o}$:

$$\sum_{s,a} \sum_{k=1}^{m} \nu_k(s,a) \left(\widetilde{r}_k(s,a) - \overline{r}(s,a)\right) \leq \square r_{\max} \sqrt{SAT \log \left( \frac{SAT}{\delta} \right)}$$

where $SA$ is actually the cardinal of the set $\bigcup_{o\in\mathcal{O}}\bigcup_{s\in\mathcal{S}_o}\{\pi_o(s)\}$, which is upper-bounded by the true number of state-action pairs in the original MDP.

The third term of (36) is a martingale difference sequence:

$$\sum_{s,o}\sum_{k=1}^{m}\omega_k^+(s,o)\left(t_k(s,o)-\nu_k(s,o)\overline{\tau}(s,o)\right)=\sum_{k=1}^{m}\sum_{i=i_k}^{i_{k+1}-1}\omega_k^+(s_i,o_i)\left(\tau_i(s_i,o_i)-\overline{\tau}(s_i,o_i)\right)$$

Denoting $X_i=\omega_k^+(s_i,o_i)\left(\tau_i(s_i,o_i)-\overline{\tau}(s_i,o_i)\right)$ and $\mathcal{F}_i=\sigma\left(s_1,o_1,R_1,\tau_1,...,s_i,o_i\right)$ the sigma algebra generated by the sequence of states, options, rewards and durations. We have that $\mathbb{E}\left[X_{i+1}|\mathcal{F}_i\right]=0$ so the above sum is indeed a martingale difference sequence.

**Theorem 4** (Wainwright [30]). *Let $(X_i,\mathcal{F}_i)_i$ be a martingale difference sequence and suppose that for any $|\lambda|<1/b_i$ we have $\mathbb{E}\left[e^{\lambda X_i}|\mathcal{F}_{i-1}\right]\leq e^{\lambda^2\sigma_i^2/2}$ almost surely. Then:*

$$\forall\epsilon\geq 0,\ \mathbb{P}\left(\left|\sum_{i=1}^{n}X_i\geq\epsilon\right|\right)\leq\begin{cases}2\exp\left(\frac{-\epsilon^2}{2\sum_i^n\sigma_i^2}\right)\ if\ 0\leq\epsilon\leq\frac{\sum_i^n\sigma_i^2}{\max_i\{b_i\}}\\2\exp\left(\frac{-\epsilon}{2\max_i\{b_i\}}\right)\ otherwise\end{cases}$$

We know from Fruit and Lazaric [14] that $(\tau_i(s,o))_i$ are sub-exponential random variables and the conditions of Theorem 4 are satisfied with $\sigma_i=r_{\max}\sigma_\tau$ and $b_i=r_{\max}b_\tau$ (the factor $r_{\max}$ is coming from the fact that $\omega_k^+(s_i,o_i)\leq r_{\max}$). We thus have that there exist an event $\Omega_4$ with probability $1-\delta$ for which:

$$\forall n\geq 0,\ \sum_{s,o}\sum_{k=1}^{m}\omega_k^+(s,o)\left(t_k(s,o)-\nu_k(s,o)\overline{\tau}(s,o)\right)\leq\square r_{\max}\left(b_\tau\log\left(\frac{b_\tau\log\left(\frac{n}{\delta}\right)}{\delta\sigma_\tau}\right)+\sigma_\tau\sqrt{n\log\left(\frac{n}{\delta}\right)}\right)$$

Now we will bound the second term in (36). Since $\widetilde{\mu}_{s,o}^k$ and $\mu_{s,o}$ are probability distributions, the difference $\widetilde{\omega}_k(s,o)-\omega_k^+(s,o)$ is not impacted if a constant is added to all terms of $\left(\widetilde{r}_{s,o}^k(s')\right)_{s'\in\mathcal{S}_o}$. In particular we can subtract the term $\left(\max_{s'\in\mathcal{S}_o}\{\widetilde{r}_{s,o}^k(s')\}+\min_{s'\in\mathcal{S}_o}\{\widetilde{r}_{s,o}^k(s')\}\right)/2$ and we obtain[11]:

$$\widetilde{\omega}_k(s,o)-\omega_k^+(s,o)\leq\frac{1}{2}\operatorname{sp}\{\widetilde{r}_{s,o}^k\}\|\widetilde{\mu}_{s,o}^k-\mu_{s,o}\|_1\leq\frac{1}{2}\times\kappa_{s,o}^1\times\operatorname{sp}\{\widetilde{r}_{s,o}^k\}\times\|\widetilde{P}_{s,o}^k-P_{s,o}\|_{\infty,1}\quad(37)$$

where $\|.\|_{\infty,1}$ for a matrix corresponds to the maximum $\ell_1$ norm of the rows. Due to the confidence intervals that we use for transition probabilities, the $\ell_1$ deviation of the empirical distribution is bounded as follows (with probability at least $1-\delta$):

$$\|P_{s,o}-\widehat{P}_{s,o}^k\|_{\infty,1}\leq\square\left(\frac{S_o\log\left(\frac{S_o\log(T_k(s,o))}{\delta}\right)}{\min_{s'\in\mathcal{S}_o}\{N_o^k(s')\}}+\sqrt{\frac{|\operatorname{supp}\{\widehat{P}_{s,o}\}|-1}{\min_{s'\in\mathcal{S}_o}\{N_o^k(s')\}}\log\left(\frac{S_o\log\left(T_k(s,o)\right)}{\delta}\right)}\right)$$

where $|\operatorname{supp}\{\widehat{P}_{s,o}\}|\leq S_o$ is the maximum support of the rows of $\widehat{P}_{s,o}$. The above bound is obtained by summing the bounds of all the terms of every row probability vector and applying Cauchy-Schwartz inequality: $\sum_{x=1}^{X}\sqrt{p_x(1-p_x)}\leq\sqrt{\left(\sum_{x=1}^{X}p_x\right)\left(X-\sum_{x=1}^{X}p_x\right)}=\sqrt{X-1}$. Since the transitions outside the support are never observed (they have a zero probability of occurrence by definition of the support) we also have: $|\operatorname{supp}\{\widehat{P}_{s,o}\}|\leq|\operatorname{supp}\{P_{s,o}\}|\leq\max_{s,o}\{|\operatorname{supp}\{P_{s,o}\}|\}=B^*$. Since $k\notin\bigcup_{s,o}K_{s,o}$ we have: $1/\min_{s'\in\mathcal{S}_o}\{N_o^k(s')\}\leq 2/(\mu_{s,o}^\star T_k(s,o))$. Moreover since $M\in\mathcal{M}_k$, $\|\widetilde{P}_{s,o}^k-P_{s,o}\|_{\infty,1}$ is bounded by twice the above bound on $\|P_{s,o}-\widehat{P}_{s,o}^k\|_{\infty,1}$. Under event $\Omega_3$ we have that:

$$\forall s,o,s',\forall k,\ |\widetilde{r}_{s,o}^k(s')-\overline{r}_{s,o}(s')|\leq\square r_{\max}\sqrt{\frac{\log\left(SAT/\delta\right)}{N_o^k(s')}}$$

$$\Rightarrow\ \forall s,o,\forall k,\ \operatorname{sp}\{\widetilde{r}_{s,o}^k\}-\operatorname{sp}\{\overline{r}_{s,o}\}\leq\operatorname{sp}\{\widetilde{r}_{s,o}^k-\overline{r}_{s,o}\}\leq 2\|\widetilde{r}_{s,o}^k-\overline{r}_{s,o}\|_\infty$$

$$\leq\square r_{\max}\sqrt{\frac{\log\left(SAT/\delta\right)}{\min_{s'\in\mathcal{S}_o}\{N_o^k(s')\}}}$$

By adjusting $\delta$ and taking a union bound over $s$, $o$ and $T$ we can create an event $\Omega_5$ of probability at least $1 - \delta$ for which:

$$\sum_{s,o}\sum_{k=1}^{m} t_k(s,o)\left(\widetilde{\omega}_k(s,o) - \omega_k^+(s,o)\right) \leq \square \sum_{s,o}\sum_{k=1}^{m} \frac{t_k(s,o)}{T_k(s,o)\mu^\star}\, \mathrm{sp}\{\widetilde{r}_{s,o}^k\}\kappa_{s,o}^1 S_o \log\left(\frac{S_o S' OT \log(T)}{\delta}\right)$$

$$+\square \sum_{s,o}\sum_{k=1}^{m} \frac{t_k(s,o)}{T_k(s,o)} r_{\max}\kappa_{s,o}^1 \sqrt{\frac{B^* - 1}{\mu^\star} \log\left(\frac{S_o S' OT \log(T)}{\delta}\right)}$$

$$+\square \sum_{s,o}\sum_{k=1}^{m} \frac{t_k(s,o)}{\sqrt{T_k(s,o)}}\, \mathrm{sp}\{\overline{r}_{s,o}\}\kappa_{s,o}^1 \sqrt{\frac{B^* - 1}{\mu^\star} \log\left(\frac{S_o S' OT \log(T)}{\delta}\right)}$$

$$\leq \square r_{\max}\frac{\kappa_\star^1}{\mu^\star} m SS' O \log\left(\frac{SS' OT \log(T)}{\delta}\right)$$

$$+\square r_{\max}\kappa_\star^1 \sqrt{\frac{B^\star - 1}{\mu^\star} \log\left(\frac{SS' OT \log(T)}{\delta}\right)}$$

$$+\square r^\star \kappa_\star^1 \sqrt{\frac{B^\star - 1}{\mu^\star}} S' OT \log\left(\frac{SS' OT \log(T)}{\delta}\right)$$

where $\kappa_\star^1 = \max_{s,o}\{\kappa_{s,o}^1\}$, $r^\star = \max_{s,o}\{\overline{r}_{s,o}\}$ and $\mu^\star = \min_{s,o}\{\mu_{s,o}^\star\}$. Here we used the fact that due to the stopping condition of an episode: $t_k(s,o) \leq T_k(s,o)$ and $\sum_{s,o}\sum_{k=1}^{m} \frac{t_k(s,o)}{\sqrt{T_k(s,o)}} \leq \square\sqrt{SOT}$ (see Lemma 19 of Jaksch et al. [15]). Furthermore, the number of episodes is only logarithmic in $T$: $m \leq \square SA \log\left(\frac{T}{SA}\right)$.

Finally, we need to bound the first term of (36):

$$\widetilde{\Delta}_k = \sum_{s,o} t_k(s,o)\left(\rho^* - \widetilde{\omega}_k(s,o)\right) \leq \sum_{s,o} t_k(s,o)\left(\widetilde{\rho}_k - \widetilde{\omega}_k(s,o)\right) + r_{\max}\sum_{s,o}\frac{t_k(s,o)}{\sqrt{t_k}}$$

$$\leq \sum_{s,o} t_k(s,o)\left(\widetilde{\rho}_k - \widetilde{\omega}_k(s,o)\right) + r_{\max}\sum_{s,o}\frac{t_k(s,o)}{\sqrt{T_k(s,o)}}$$

We further decompose the remaining term:

$$\sum_{s,o} t_k(s,o)\left(\widetilde{\rho}_k - \widetilde{\omega}_k(s,o)\right) = \sum_{s,o} t_k(s,o)\left(\widetilde{\rho}_k - \widetilde{\omega}_k(s,o)\right)\left(\mathbb{1}_{\{\widetilde{\tau}_k(s,o)<+\infty\}} + \mathbb{1}_{\{\widetilde{\tau}_k(s,o)=+\infty\}}\right)$$

$$= \sum_{s,o} \nu_k(s,o)\widetilde{\tau}_k(s,o)\left(\widetilde{\rho}_k - \widetilde{\omega}_k(s,o)\right)\mathbb{1}_{\{\widetilde{\tau}_k(s,o)<+\infty\}}$$

$$+ \sum_{s,o}\left(t_k(s,o) - \nu_k(s,o)\widetilde{\tau}_k(s,o)\right)\left(\widetilde{\rho}_k - \widetilde{\omega}_k(s,o)\right)\mathbb{1}_{\{\widetilde{\tau}_k(s,o)<+\infty\}}$$

$$+ \sum_{s,o} t_k(s,o)\left(\widetilde{\rho}_k - \widetilde{\omega}_k(s,o)\right)\mathbb{1}_{\{\widetilde{\tau}_k(s,o)=+\infty\}}$$

$$(38)$$

When $\widetilde{\tau}_k(s,o) = +\infty$ i.e., $\widetilde{\mu}_{s,o}^k(s) = 0$, using the fact that $\forall S'$, $|u_{j+1}(s) - u_j(s) - \widetilde{\rho}_k| \leq r_{\max}/\sqrt{t_k}$ and using (7) we obtain: $|\widetilde{\rho}_k - \widetilde{\omega}_k(s,o)| \leq r_{\max}/\sqrt{t_k}$. So the last term of (38) is bounded as follows:

$$\sum_{s,o} t_k(s,o)\left(\widetilde{\rho}_k - \widetilde{\omega}_k(s,o)\right)\mathbb{1}_{\{\widetilde{\tau}_k(s,o)=+\infty\}} \leq r_{\max}\sum_{s,o}\frac{t_k(s,o)}{\sqrt{t_k}}\mathbb{1}_{\{\widetilde{\tau}_k(s,o)=+\infty\}} \leq r_{\max}\sum_{s,o}\frac{t_k(s,o)}{\sqrt{T_k(s,o)}}$$

We can bound the first term of (38) using again $\forall S'$, $|u_{j+1}(s) - u_j(s) - \widetilde{\rho}_k| \leq r_{\max}/\sqrt{t_k}$:

$$\sum_{s,o} \nu_k(s,o)\widetilde{\tau}_k(s,o)\left(\widetilde{\rho}_k - \widetilde{\omega}_k(s,o)\right)\mathbb{1}_{\{\widetilde{\tau}_k(s,o)<+\infty\}} \leq \boldsymbol{\nu}_k^\intercal\left(\widetilde{B}_k - I\right)\boldsymbol{w}_k\mathbb{1}_{\{\widetilde{\tau}_k(s,o)<+\infty\}} + r_{\max}\sum_{s,o}\frac{t_k(s,o)}{\sqrt{T_k(s,o)}}$$

where $\widetilde{B}_k$ is the optimistic transition matrix of the SMDP under the greedy policy $\widetilde{\pi}_k$ of EVI (7), $\boldsymbol{\nu}_k = \left(\nu_k(s,\widetilde{\pi}_k(s))\right)_{s\in\mathcal{S}'}$ and $\boldsymbol{w}_k$ corresponds to $\boldsymbol{u}_j - \frac{1}{2}\left(\max\{\boldsymbol{u}_j\} + \min\{\boldsymbol{u}_j\}\right)\boldsymbol{e}$. The term

$\boldsymbol{\nu}_{\boldsymbol{k}}^{\mathsf{T}} \left( \widetilde{B}_k - I \right) \boldsymbol{w_k}$ was analysed by Jaksch et al. [15] (MDP) and Fruit and Lazaric [14] (SMDP) for Hoeffding confidence intervals on the transition probabilities. Here we are using Bernstein confidence bounds like Dann and Brunskill [19] so we have:

$$
\sum_{k=1}^{m} \boldsymbol{\nu}_{\boldsymbol{k}}^{\mathsf{T}} \left( \widetilde{B}_k - I \right) \boldsymbol{w_k} \mathbb{1}_{\{\widetilde{\tau}_k(s,o)<+\infty\}} \leq \Box D' \sqrt{(B'-1)S'On \log\left(\frac{n}{\delta}\right)} + \Box D' \sqrt{n \log\left(\frac{n}{\delta}\right)}
$$
$$
+ \Box D' SA \log\left(\frac{T}{SA}\right)
$$
$$
\leq \Box D' \sqrt{B'S'On \log\left(\frac{n}{\delta}\right)} + \Box D' SA \log\left(\frac{T}{SA}\right)
$$

where $D'$ is the diameter of the SMDP induced by options and $B' = \max_{s,o}\{|\mathrm{supp}\{\boldsymbol{b_{s,o}}\}|\}$ is the maximal support of transition probability vectors in the SMDP.

Finally, the second term of (38) can be decomposed as follows:

$$
\sum_{s,o} \left( t_k(s,o) - \nu_k(s,o)\widetilde{\tau}_k(s,o) \right) \left( \widetilde{\rho}_k - \widetilde{\omega}_k(s,o) \right) \mathbb{1}_{\{\widetilde{\tau}_k(s,o)<+\infty\}} =
$$
$$
\sum_{s,o} \left( t_k(s,o) - \nu_k(s,o)\overline{\tau}(s,o) \right) \left( \widetilde{\rho}_k - \widetilde{\omega}_k(s,o) \right) \mathbb{1}_{\{\widetilde{\tau}_k(s,o)<+\infty\}} \qquad (39)
$$
$$
+ \sum_{s,o} \nu_k(s,o) \left( \overline{\tau}(s,o) - \widetilde{\tau}_k(s,o) \right) \left( \widetilde{\rho}_k - \widetilde{\omega}_k(s,o) \right) \mathbb{1}_{\{\widetilde{\tau}_k(s,o)<+\infty\}}
$$

The first term of (39) is similar to the third term of (36): it is a martingale difference sequence and under event $\Omega_4$ we have:

$$
\forall n \geq 0, \ \sum_{s,o} \sum_{k=1}^{m} \left( t_k(s,o) - \nu_k(s,o)\overline{\tau}(s,o) \right) \left( \widetilde{\rho}_k - \widetilde{\omega}_k(s,o) \right) \mathbb{1}_{\{\widetilde{\tau}_k(s,o)<+\infty\}}
$$
$$
\leq \Box r_{\max} \left( b_\tau \log\left(\frac{b_\tau \log\left(\frac{n}{\delta}\right)}{\delta \sigma_\tau}\right) + \sigma_\tau \sqrt{n \log\left(\frac{n}{\delta}\right)} \right)
$$

If $\overline{\tau}(s,o) \geq \widetilde{\tau}_k(s,o)$ then $\widetilde{\mu}_{s,o}^k(s) - \mu_{s,o}(s) \geq 0$ and $\overline{\tau}(s,o) - \widetilde{\tau}_k(s,o) \leq \overline{\tau}(s,o)^2 \left( \widetilde{\mu}_{s,o}^k(s) - \mu_{s,o}(s) \right)$ implying that:

$$
\left( \overline{\tau}(s,o) - \widetilde{\tau}_k(s,o) \right) \left( \widetilde{\rho}_k - \widetilde{\omega}_k(s,o) \right) \mathbb{1}_{\{\overline{\tau}_k(s,o)\geq\widetilde{\tau}_k(s,o)\}} \leq r_{\max}\overline{\tau}(s,o)^2 \times \|\widetilde{\mu}_{s,o}^k - \mu_{s,o}\|_\infty
$$
$$
\leq r_{\max}\overline{\tau}(s,o)^2 \times \kappa_{s,o}^\infty \times \|\widetilde{P}_{s,o}^k - P_{s,o}\|_{\infty,1}
$$

We already gave an upper bound for the term $\|\widetilde{P}_{s,o}^k - P_{s,o}\|_{\infty,1}$.

$$
\sum_{k=1}^{m} \sum_{s,o} \nu_k(s,o) \left( \overline{\tau}(s,o) - \widetilde{\tau}_k(s,o) \right) \left( \widetilde{\rho}_k - \widetilde{\omega}_k(s,o) \right) \mathbb{1}_{\{\overline{\tau}_k(s,o)\geq\widetilde{\tau}_k(s,o)\}}
$$
$$
\leq r_{\max} \sum_{k=1}^{m} \sum_{s,o} \nu_k(s,o)\overline{\tau}(s,o)^2 \kappa_{s,o}^\infty \|\widetilde{P}_{s,o}^k - P_{s,o}\|_{\infty,1}
$$

and:

$$
\sum_{k=1}^{m} \sum_{s,o} \nu_k(s,o)\overline{\tau}(s,o)^2 \kappa_{s,o}^\infty \|\widetilde{P}_{s,o}^k - P_{s,o}\|_{\infty,1} = \sum_{k=1}^{m} \sum_{s,o} t_k(s,o)\overline{\tau}(s,o) \kappa_{s,o}^\infty \|\widetilde{P}_{s,o}^k - P_{s,o}\|_{\infty,1}
$$
$$
+ \sum_{k=1}^{m} \sum_{s,o} \left( \nu_k(s,o)\overline{\tau}(s,o) - t_k(s,o) \right) \overline{\tau}(s,o) \kappa_{s,o}^\infty \|\widetilde{P}_{s,o}^k - P_{s,o}\|_{\infty,1}
$$

Once again, the last term is a martingale difference sequence so we can apply Theorem 4. Under events $\Omega_4$ and $\Omega_5$:

$$\sum_{k=1}^{m}\sum_{s,o}\left(\nu_k(s,o)\overline{\tau}(s,o)-t_k(s,o)\right)\overline{\tau}(s,o)\kappa_{s,o}^{\infty}\|\widetilde{P}_{s,o}^{k}-P_{s,o}\|_1$$

$$=\sum_{k=1}^{m}\sum_{i=i_k}^{i_{k+1}-1}\overline{\tau}(s_i,o_i)\kappa_{s_i,o_i}^{\infty}\|\widetilde{P}_{s_i,o_i}^{k}-P_{s_i,o_i}\|_1\left(\overline{\tau}(s_i,o_i)-\tau_i(s_i,o_i)\right)$$

$$\leq\begin{cases}\square\tau_{\max}\kappa_{\star}^{\infty}b_{\tau}\log\left(\frac{n}{\delta}\right) & \text{if } ...\\[2mm]\square\tau_{\max}\kappa_{\star}^{\infty}\sigma_{\tau}\sqrt{\log\left(\frac{n}{\delta}\right)\sum_{s,o}\sum_{k=1}^{m}\nu_k(s,o)\left[\frac{S_o\log(...)}{T_k(s,o)\mu^{\star}}+\sqrt{\frac{(B^{\star}-1)\log(...)}{\mu^{\star}T_k(s,o)}}\right]^2} & \text{otherwise}\end{cases}$$

where $\kappa_{\star}^{\infty}=\max_{s,o}\{\kappa_{s,o}^{\infty}\}$. For the sake of clarity and brevity, we do not detail the above bounds any further: since $T_k(s,o)\geq N_k(s,o)$ and $\nu_k(s,o)\leq N_k(s,o)$, it is clear that it is bounded by a logarithmic term which will be ignored for the final bound. Under event $\Omega_5$, the final term can be bounded as follows:

$$\sum_{k=1}^{m}\sum_{s,o}t_k(s,o)\overline{\tau}(s,o)\kappa_{s,o}^{\infty}\|\widetilde{P}_{s,o}^{k}-P_{s,o}\|_1\leq\square\tau_{\max}\kappa_{\star}^{\infty}\sum_{s,o}\sum_{k=1}^{m}\frac{t_k(s,o)}{T_k(s,o)\mu^{\star}}S_o\log\left(\frac{S_oS'OT\log(T)}{\delta}\right)$$

$$+\square\tau_{\max}\kappa_{\star}^{\infty}\sum_{s,o}\sum_{k=1}^{m}\frac{t_k(s,o)}{\sqrt{T_k(s,o)}}\sqrt{\frac{B^{*}-1}{\mu^{\star}}\log\left(\frac{S_oS'OT\log(T)}{\delta}\right)}$$

$$\leq\square\tau_{\max}\frac{\kappa_{\star}^{\infty}}{\mu^{\star}}mSS'O\log\left(\frac{S_oS'OT\log(T)}{\delta}\right)$$

$$+\square\tau_{\max}\kappa_{\star}^{\infty}\sum_{s,o}\sqrt{\frac{B^{*}-1}{\mu^{\star}}S'OT\log\left(\frac{S_oS'OT\log(T)}{\delta}\right)}$$

## D.7 Gathering all the terms

If we adjust $\delta$, take a union bound over $\Omega_1,...,\Omega_5$ and ignore all logarithmic terms, we find the bound of Theorem 1:

$$\Delta=\widetilde{O}\left(D'\sqrt{B'S'On}+(\sigma_R+r_{\max}\sigma_{\tau})\sqrt{n}+r_{\max}\sqrt{SAT}+(r^{*}\kappa_{*}^{1}+r_{\max}\tau_{\max}\kappa_{*}^{\infty})\sqrt{\frac{B^{*}}{\mu^{*}}S'OT}\right)$$

## D.8 Relaxing preliminary assumptions

We now show how Asm. 3 can be relaxed without impacting the main terms of the regret bound.

### D.8.1 Approximate nested EVI

We assumed that the exact version of EVI (7) is run, and not the nested EVI algorithm presented in the article. Unfortunately, the exact EVI is not runnable in practice because (11) requires an infinite number of iterations to converge in the general case. However, Thm. 2 of App. C shows that all the guarantees of the exact algorithm are preserved with the approximate one. More precisely, the algorithm converges in span semi-norm and we can tune the stopping condition so as to enforce that the optimality equation is satisfied with a given level of accuracy $\varepsilon$. Finally, it is possible to tune the algorithm to guarantee that the span of the final value function is bounded by $D'+1$ which is of the order of $D'$ ($D'\geq 1$ by definition). To do so we should tune $(\varepsilon_j)_{j\geq 0}$ such that $\sum_{j\geq 0}\varepsilon_j\leq 1$. These properties are all we need to derive the main term $\widetilde{O}\left(D'\sqrt{B'S'On}\right)$ in the regret bound.

### D.8.2 Stopping condition of an episode

We assumed that the number of visits in a state-action pair of the MDP can only occur at the end of the execution of an option. This is not the case in general: once the number of visits has doubled in

one state-action pair, the algorithm needs to wait for the option being executed to end before starting a new episode. On average, ending the option might take up to $\tau_{\max}$ time steps (with a variance of $\sigma_\tau$). This can only decrease the bound on the number of episodes $m$ for a given time horizon $T$. So we will still have: $m \leq \square SA \log\left(\frac{T}{SA}\right)$. On the other hand, the inequality $t_k(s,o) \leq T_k(s,o)$ is no longer satisfied and should be replaced by: $t_k(s,o) \leq T_k(s,o) + \tau_{\max} + \square\sigma_\tau\sqrt{\log\left(\frac{S'OT}{\delta}\right)}$ (in high probability). This will have only a minor impact on the regret by introducing additional logarithmic terms. Namely, instead of having $\sum_{s,o}\sum_{k=1}^{m}\frac{t_k(s,o)}{T_k(s,o)} \leq 1$ we have:

$$\sum_{s,o}\sum_{k=1}^{m}\frac{t_k(s,o)}{T_k(s,o)} \leq 1 + \sum_{s,o}\sum_{k=1}^{m}\frac{\tau_{\max} + \square\sigma_\tau\sqrt{\log\left(\frac{S'OT}{\delta}\right)}}{T_k(s,o)}$$

where the second term is logarithmic in $T$ (since $m$ is logarithmic). Moreover, the proof of Lem. 19 of Jaksch et al. [15] can also be adapted to show that $\sum_{s,o}\sum_{k=1}^{m}\frac{t_k(s,o)}{\sqrt{T_k(s,o)}}$ is bounded by $\square\sqrt{S'OT}$ plus a logarithmic term:

**Lemma 5.** *For any non-negative constant $C \geq 0$ and any sequence of numbers $z_1, \ldots, z_n$ with $z_k \leq Z_{k-1} + C$ and $\leq Z_{k-1} = \max\left\{1, \sum_{i=1}^{k-1}z_i\right\}$ we have:*

$$\sum_{k=1}^{n}\frac{z_k}{\sqrt{Z_{k-1}}} \leq \frac{\sqrt{2}}{\sqrt{2}-1}\sqrt{Z_n} + nC$$

*Proof.* Since $z_k \leq Z_{k-1} + C$ we can write:

$$\sum_{k=1}^{n}\frac{z_k}{\sqrt{Z_{k-1}}} \leq \sum_{k=1}^{n}\sqrt{Z_{k-1}} + C\sum_{k=1}^{n}\underbrace{\frac{1}{\sqrt{Z_{k-1}}}}_{\leq 1} \leq \sum_{k=1}^{n}\sqrt{Z_{k-1}} + nC$$

The term $\sum_{k=1}^{n}\sqrt{Z_{k-1}}$ is maximal when $z_k = Z_{k-1} + C$ for all $k \in \{1, \ldots, n\}$ in which case we can prove (after solving the induction):

$$Z_0 = 1 \text{ and } Z_k = 2^{k-1} + (2^k - 1)\cdot C \leq 2^{k-1}\cdot(1+2C), \forall k \geq 1$$

Therefore:

$$\sum_{k=1}^{n}\sqrt{Z_{k-1}} \leq 1 + \sqrt{1+2C}\cdot\sum_{k=2}^{n}\left(\sqrt{2}\right)^{k-2} = 1 + \sqrt{1+2C}\cdot\frac{\left(\sqrt{2}\right)^{n-1}-1}{\sqrt{2}-1}$$

$$\leq 1 + \frac{\sqrt{2^{n-1}+2^nC}}{\sqrt{2}-1} = 1 + \frac{\sqrt{Z_n}}{\sqrt{2}-1}$$

$$\leq \underbrace{\sqrt{Z_n}}_{1\leq Z_n} + \frac{\sqrt{Z_n}}{\sqrt{2}-1} = \frac{\sqrt{2}}{\sqrt{2}-1}\sqrt{Z_n}$$

$\square$

Compared to Lem. 19 of Jaksch et al. [15], we have the additional term $nC$ in the bound. In our case, $C = \tau_{\max} + \square\sigma_\tau\sqrt{\log\left(\frac{S'OT}{\delta}\right)}$ and $n = m \leq \square SA\log\left(\frac{T}{SA}\right)$ so the additional term is indeed logarithmic.

### D.8.3   Aperiodicity of options

In order to be able to approximate $N_o^k(s')$ by $T_k(s,o)\mu_{s,o}(s')$ in the proof, we used a Bernstein inequality for ergodic Markov Chains (Thm. 4). This inequality only holds when the chain is aperiodic because otherwise the pseudo-spectral gap is not defined. To overcome this problem, we can use the so-called "aperiodicity transformation" of a Markov chain (see e.g., [27]) that consists in adding a self-loop with equal probability $0 < \alpha < 1$ in every state, thus making the chain aperiodic

while preserving the stationary distribution. More formally, the transition matrix $P_\alpha$ of the new chain is a convex combination of the transition matrix of the original chain $P$ and the identity matrix: $P_\alpha \longleftarrow (1-\alpha)P + \alpha I$. It is trivial to see that if $P$ is irreducible with stationary distribution $\mu$ than $P_\alpha$ is also irreducible with stationary distribution $\mu_\alpha = \mu$ and is in addition aperiodic (with spectral gap $\gamma_\alpha$.

Assume we have $n$ samples $X_1...X_n$ drawn from an irreducible periodic Markov Chain $P$. For all $i = 1,...,n$ we keep sampling a Bernouilli $\mathcal{B}(\alpha)$ until a 0 is obtained. For all 1s that were observed we duplicate the sample $X_i$. The process obtained in this way is denoted $X_1^\alpha,...X_{n+n_\alpha}^\alpha$ where $n_\alpha$ is the random variable corresponding to the number of additional samples. Conditionally on knowing $n + n_\alpha$, $X_1^\alpha,...X_{n+n_\alpha}^\alpha$ is distributed as a Markov Chain $P_\alpha$. Therefore, using Thm. 3 we have that with high probability:

$$\left| \sum_{i=1}^{n+n_\alpha} f(X_i^\alpha) - (n+n_\alpha)\mathbb{E}_\mu[f(X)] \right| = \widetilde{O}\left( \sqrt{\frac{V_f}{\gamma_\alpha}(n+n_\alpha)} \right)$$

For all $i = 1,...,n$, denote by $\tau_i(\alpha)$ the (random) number of consecutive 1s before the first 0 is observed when sequentially sampling i.i.d. Bernouilli distributions $\mathcal{B}(\alpha)$. The probability of first observing a 0 after $k$ samples is $\alpha^k(1-\alpha)$ meaning that $\tau_i(\alpha)$ has a geometric distribution with parameter $1-\alpha$ and $\mathbb{E}[\tau_i(\alpha)] = \alpha/(1-\alpha)$. It is well-known from the literature that any geometric distribution is sub-Exponential so we have that with high probability:

$$\left| n_\alpha - \frac{\alpha}{1-\alpha}n \right| = \left| \sum_{i=1}^{n} \tau_i(\alpha) - \mathbb{E}\left[ \sum_{i=1}^{n} \tau_i(\alpha) \right] \right| = \widetilde{O}\left( \sqrt{\frac{\alpha}{(1-\alpha)^2}n} \right)$$

Combining the two results and using the fact that $\alpha \leq 1$ and for any integer $n$, $\sqrt{n} \leq n$ we have with high probability:

$$\left| \sum_{i=1}^{n+n_\alpha} f(X_i^\alpha) - (n+n_\alpha)\mathbb{E}_\mu[f(X)] \right| = \widetilde{O}\left( \sqrt{\frac{V_f}{(1-\alpha)\gamma_\alpha}n} \right)$$

So in conclusion, when the chain is not aperiodic, we can apply the aperiodicity transformation to obtain the same kind of concentration inequality with $(1-\alpha)\gamma_\alpha$ instead of $\gamma$.

## E  Detailed example of option

In order to better understand the terms $\Delta_\mu$ and $\Delta'_{R,\tau}$ appearing in the regret bound, we consider a very simple option represented on Fig. 5. Although this option is the simplest that we can think of (only two inner states), it is sufficiently expressive to get a good intuition of what is happening in the general case.

Figure 5: Illustrative example of option: the starting state of the option is $s_0$, $s_1$ is an inner state with $\beta(s_1) = 0$ (the option never stops in $s_1$) and $s_2$ is a terminal state (i.e, $\beta(s_2) = 1$). We assume that $p_1 > 0$ so that the option is well-defined.

## E.1 Sub-Exponential constants (regret term $\Delta'_{R,\tau}$)

The expected duration of the option is $\overline{\tau} = 1 + p_0/p_1$ while the expected reward is $\overline{R} = (p_0/p_1)R_{\max}$. For the variance, we have a simple closed form formula $\sigma_\tau^2 = (p_0/p_1^2)(2 - p_0 - p_1)$ and we observed empirically that $b_\tau = 1/p_1$ is a valid (and tight) sub-exponential constant [12]. This example illustrates why an option can be very difficult to learn. If we assume for example that $p_1 << p_0 << 1$, the sub-exponential constants become very big. The intuitive reason for this is that state $s_1$ has a huge impact on the average duration and reward but is reached with low probability. Thus, a lot of samples are required to learn the option accurately (i.e., get samples from $s_1$). This cannot happen if the option does not contain any cycle. Actually, Fruit and Lazaric [14] proved that the duration and reward of an option can either be sub-exponential when there exists cycles or bounded when there is no cycle (in particular bounded is equivalent to sub-Gaussian).

## E.2 Pseudo-diameter of the option (regret term $\Delta_\mu$)

We give the closed-form formulas of some condition numbers and the stationary distribution without detailing the calculations [13]:

$$\boldsymbol{\mu} = \frac{1}{p_0 + p_1}[p_1, p_0]^\mathsf{T}, \quad \kappa^1 = \tau_1(H) = \frac{1}{p_0 + p_1}, \quad \kappa^\infty = \frac{1}{2}\max_s\left\{\frac{\max_{s' \neq s} m_{s',s}}{m_{s,s}}\right\} = \frac{1}{2(p_0 + p_1)}$$

The pseudo-diameter of the option is thus:

$$\widetilde{D} = \frac{r_{\max}}{\sqrt{(p_0 + p_1)\min\{p_o, p_1\}}} + \frac{r_{\max}}{2p_1}\sqrt{\frac{p_0 + p_1}{\min\{p_o, p_1\}}}$$

After analysing all the possible configurations ($p_0 \leq p_1$, $p_0 > p_1$), we find that we always have $\widetilde{D} \geq r_{\max}\sigma_\tau$. Moreover, $\Delta'_{R,\tau}$ scales as $\sqrt{n}$ while $\Delta_\mu$ scales as $\sqrt{T}$ so $\Delta_\mu$ is always significantly worse than $\Delta'_{R,\tau}$. This seems to be the price to pay for removing all prior knowledge on the parameters of the option. Note however that our regret analysis is very worst-case (the condition numbers might not always be tight). Moreover, the correlation between options, the span of the internal rewards or the support of the inner transition probabilities (within an option) can all reduce the value of $\Delta_\mu$.

## E.3 Interpreting sub-exponential constants using the irreducible chain view

The irreducible Markov chain corresponding to the option of Fig. 5 is always aperiodic and reversible. The spectral gap is $\gamma = p_0 + p_1$. We note that when $p_0 + p_1 << 2$, $\sigma_\tau$ is of the order of $\sqrt{p_0}/p_1$ which corresponds to $\sqrt{\overline{\tau}(\overline{\tau} - 1)/\gamma}$. On the other hand, $\beta_\tau$ is of the order of $1/p_1$ which corresponds to $(\overline{\tau} - 1)/\gamma$. Actually this correspondence between the sub-exponential constants $(b_\tau, \sigma_\tau)$ and $\gamma$ and $\overline{\tau}$ can be explained by the fact that the terms $\sigma_\tau$ appearing in our new regret bound comes from the term $t_k(s, o) - \nu_k(s, o)\overline{\tau}(s, o)$ that we bounded using Azuma's inequality for sub-exponential random variables (Thm. 3). But we could also note that

$$t_k(s, o) - \nu_k(s, o)\overline{\tau}(s, o) = \overline{\tau}(s, o)(\mu_{s,o}(s)t_k(s, o) - \nu_k(s, o))$$

where $\mu_{s,o}(s)t_k(s, o) - \nu_k(s, o)$ has the form $n\mathbb{E}_\mu[f(X)] - \sum_{i=1}^n f(X_i)$ and $(X_i)_{1 \leq i \leq n}$ is the sequence of visited states in the ergodic Markov Chain representing the option and $f(X_i) = \mathbb{1}_{\{X_i = s\}}$. As we did in App. D.5, we can use Bernstein inequality for Markov chains (Thm. 3) to show:

$$\mu_{s,o}(s)t_k(s, o) - \nu_k(s, o) \leq \Box\sqrt{\frac{\mu_{s,o}(s)(1 - \mu_{s,o}(s))}{\gamma_{s,o}}t_k(s, o)\log\left(\frac{\sqrt{\overline{\tau}(s, o)}}{\delta}\right)}$$

$$+ \Box\frac{1}{\gamma_{s,o}}\log\left(\frac{\sqrt{\overline{\tau}(s, o)}}{\delta}\right)$$

If for all $s \in \mathcal{S}$ and $o \in \mathcal{O}$, $\gamma_{s,o} = \gamma$ and $\overline{\tau}(s,o) = \overline{\tau}$ we further have:

$$\sum_{s,o} \sum_{k=1}^{m} t_k(s,o) - \nu_k(s,o)\overline{\tau}(s,o) \leq \square \sqrt{\frac{\overline{\tau}-1}{\gamma} T \log\left(\frac{\overline{\tau}T}{\delta}\right)}$$

$$+ \square \frac{\overline{\tau}}{\gamma} \log\left(\frac{\overline{\tau}T}{\delta}\right)$$

Looking at the bound obtained and comparing it with the one derived in App. D, we clearly see the correspondence $\sigma_\tau \longleftrightarrow \sqrt{\overline{\tau}(\overline{\tau}-1)/\gamma}$ (since $T \sim n\overline{\tau}$) and $\beta_\tau \longleftrightarrow (\overline{\tau}-1)/\gamma$ appearing. Thus, interpreting an option as an irreducible Markov Chain allows us to have a better intuition about the actual meaning of the sub-exponential constants presented by Fruit and Lazaric [14]: "mixing time" of the option (of the order of $1/\gamma$), average duration, ...

# F   Experiments

This section aims at providing a detailed empirical analysis of the OFU approaches with options. Here we consider three main OFU methods: UCRL, SUCRL and FSUCRL.[14] While UCRL is a completely specified algorithm, SUCRL and FSUCRL group several approaches that differ for the amount of information required (SUCRL) or the solution method (FSUCRL). A complete overview and nomenclature are provided in Tab. 1. To compute the condition number for $\boldsymbol{\mu_o}$ we chose the smallest condition numbers in the list of Cho and Meyer [20] that is the (provably) smallest condition number for the $\ell_1$-norm [21, Th. 2.3].

**Evaluation.**   As done in the main paper we evaluate the algorithms based on the regret $\Delta(\mathfrak{A}, n)$ that can be (approximately) decomposed into three distinct terms:

$$\Delta \approx \Delta_p + \Delta_R + \Delta_\tau = \Delta_p + \Delta_{R,\tau},$$

where the first term $\Delta_p$ is the regret incurred when learning the transition kernel of the MDP while the second and third terms $\Delta_R$ and $\Delta_\tau$ are the regret incurred when learning (respectively) the reward function and the holding times. In most cases, $\Delta_p \gg \Delta_{R,\tau}$ i.e., $\Delta_p$ is the dominant term in the regret. If all options are primitive actions, the induced SMDP is simply the original MDP and $\Delta_\tau = 0$.

We denote by $\alpha_p$, $\alpha_r$ and $\alpha_\tau$ the numerical (multiplicative) coefficients used to shrink the confidence intervals of the transition kernel, the reward function and the holding times respectively[15]. Setting $\alpha_p$, $\alpha_r$, $\alpha_\tau < 1$ enables to speed-up convergence of the learning algorithms and avoid suffering from a worst-case regret (in practice the confidence intervals are very loose). By tuning these coefficients, we can also make either $\Delta_p$ or $\Delta_{R,\tau}$ dominant and analyze the impact on the regret of every algorithm. When running FSUCRL instead of SUCRL, we use $\alpha_{mc}$ (instead of $\alpha_\tau$) to shrink the confidence interval of the transition matrices of options (seen as irreducible Markov Chains, see Sec. 3.1).

To be fair, we always set $\alpha_p = \alpha_{mc}$ since they both reflect the degree of uncertainty on the transition kernel of the original MDP. We also set $\alpha_r = \alpha_\tau$ in all experiments since the cumulative reward and the duration of an option are somehow proportional.

## F.1   Simple Grid-World

We first consider the toy domain analyzed by Fruit and Lazaric [14] that was specifically designed to show the advantage of temporal abstraction. It is an instance of a 20x20 deterministic grid-world navigation problem where the 4 cardinal actions are replaced by 4 cardinal options with various

$$\sqrt{\frac{2\widehat{p}_k(s'|s,o)\left(1-\widehat{p}_k(s'|s,o)\right)c_{t_k,\delta}}{N_k(s,o)}} + \alpha_p \frac{7c_{t_k,\delta}}{3N_k(s,o)}.$$

| Family | Algorithm | Description |
|--------|-----------|-------------|
| FSUCRL | FSUCRLv1 | Uses empirical condition number and L1 confidence bound to compute optimistic stationary distributions of options explicitly |
|  | FSUCRLv2 | Uses two nested EVI to implicitly compute the optimistic stationary distribution of each option |
| SUCRL |  | **Prior Knowledge of the algorithm:** |
|  | SUCRLv1 | Maximal reward $r_{\max}$ and actual duration $T_{\max}$ |
|  | SUCRLv2 | Maximal expected duration $\tau_{\max}$, maximal variance of holding time $\sigma_\tau = \max_{o \in \mathcal{O}} \{\mathrm{Var}(\tau(o))\}$ and reward $\sigma_R = r_{\max}\sqrt{\tau_{\max} + \sigma_\tau^2}$ |
|  | SUCRLv3 | $\tau_{\max}$ and $\forall o \in \mathcal{O}$, $\sigma_\tau(o) = \mathrm{Var}(\tau(o))$ and $\sigma_R(o) = r_{\max}\sqrt{\tau(o) + \sigma_\tau(o)^2}$ |
|  | SUCRLv4 | Same as SUCRLv2 with $\sigma_R = 0$ |
|  | SUCRLv5 | Same as SUCRLv3 with $\sigma_R = 0$ |

Table 1: Detailed description of the different algorithms used for the experiments. SUCRL algorithms are enumerated according to the required level of prior knowledge (the higher the stronger). We computed $\sigma_\tau(o)$ based on the analytical formula relating $\sigma_\tau(o)$ to the dynamics of $o$. In this specific problem $\max_o\{b_R(o)\} = \max_o\{b_\tau(o)\} = 0$.

| Name | $\alpha_p$ | $\alpha_{mc}$ | $\alpha_r$ | $\alpha_\tau$ | $T_n$ |
|------|-----------|--------------|-----------|--------------|-------|
| Grid-C1 | 0.4 | 0.4 | 0.8 | 0.8 | $2 \cdot 10^9$ |
| Grid-C2 | 0.02 | 0.02 | | | $1.2 \cdot 10^8$ |

Table 2: Settings used in the gridworld experiments. Both the configurations are run with *Hoeffding* concentration inequalities.

maximal duration $T_{\max}$.[16] The optimal policy is the shortest path to a target state that triggers a random restart in the grid. The reward is zero everywhere except at the target where it is $r_{\max} = 1$. To be able to reproduce the results of Fruit and Lazaric [14], we ran our algorithms with Hoeffding confidence bounds for the $\ell_1$-deviation of the empirical distribution (implying that $B$ and $B_{\mathcal{O}}$ have no impact in our simulations).

In order to show the relevance of the difference regret components we consider two configurations: Grid-C1 and Grid-C2 (see Tab. 2). The difference resides in the shrinking coefficients: Grid-C1 is obtained with $\alpha_p = \alpha_{mc} = 0.4$ while in Grid-C2 we used $\alpha_p = \alpha_{mc} = 0.02$ (in both cases $\alpha_r = \alpha_\tau = 0.8$). On Fig. 6 we plot the value of the ratio $\mathcal{R} = \Delta(\mathfrak{A}, n)/\Delta(\mathrm{UCRL}, n)$ where $n = \max\{n : T_n \leq t\}$ and $\mathfrak{A} \in \{\mathrm{SUCRL}, \mathrm{FSUCRL}\}$ with different sets of options characterized by the maximal duration $T_{\max}$. When the ratio is smaller than 1, $\mathfrak{A}$ performs better than UCRL and conversely. The value of $n$ is big enough for all algorithms to have explored the environment extensively: for $t \geq T_n$ the regret increases only logarithmically and the value of the ratio is stable.

When comparing FSUCRL to UCRL, we empirically observe that the advantage of temporal abstraction is indeed preserved when removing the knowledge of option's characteristics. This shows that the benefit of temporal abstraction is not just a mere artifact of prior knowledge on the options: it can be achieved without any additional information w.r.t. UCRL.

**Grid-C1.** Under these settings the regret is dominated by the term $\Delta_p$ ($\Delta_p \gg \Delta_{R,\tau}$). We can see this by noting that for $T_{\max} = 1$, SUCRLv4 and v5 are equivalent to UCRL with known reward function[17] and they seem to perform almost like UCRL (the ratio is close to 1). The different versions of SUCRL are ordered by increasing amount of prior knowledge and we see that the more prior knowledge, the better (as expected). For small values of $T_{\max}$, FSUCRLv1 and v2 perform equally while for large values v1 is slightly worse. This is due to the condition number $\kappa^1$ used in the confidence bound of the stationary distributions of options (see Eq. (8)). This first experiment

Figure 6: Empirical ratio $\mathcal{R} = \Delta(\mathfrak{A}, n)/\Delta(\mathrm{UCRL}, T_n)$, with $\mathfrak{A} \in \{\mathrm{SUCRL}, \mathrm{FSUCRL}\}$ and $T_n = 2 \cdot 10^9$, as a function of the maximal length of options $T_{\max}$, when the regret term related to the estimation of the transition kernel is dominant (6a) and when the regret term related to the estimation of the reward function is dominant (6b). A detailed description of the different versions of both FSUCRL and SUCRL is reported on Table 1. We have performed 20 repetitions but there is no variability in the outcomes.

shows that FSUCRL is able to approach the performance of SUCRL with a reasonable amount of knowledge[18]

**Grid-C2.** By reducing the coefficients $\alpha_p$ and $\alpha_{mc}$ we can make the contribution of $\Delta_p$ almost negligible. In this case, the dominant term becomes $\Delta_{R,\tau}$. Even in this case, we can notice this by looking at $T_{\max} = 1$ where SUCRLv4 and v5 are able to outperform UCRL due to the knowledge of the reward function. When we decrease the knowledge provided to SUCRL we can notice that it is no more able to outperform FSUCRL and, sometimes, not even UCRL. The issue resides in the pessimistic estimation of the confidence interval for the option reward. For simplicity we consider SUCRLv1 but the reasoning extends to other versions. Since the option is considered as an atomic operator its uncertainty on the reward scales proportionally to $r_{\max} \cdot T_{\max}$. In normal settings (as in Grid-C1) this uncertainty is reduced while exploring the transition kernel but here it represents the dominating term. On contrary, FSUCRL does not show this problem since it is able to estimate the reward directly at the level of primitive actions without incurring in a penalization proportional to $T_{\max}$. As a consequence, it is able to outperform SUCRL in most of the cases.

From these experiments, we can draw comments about the presented algorithms. As expected, the more prior knowledge, the better the regret. However, unlike FSUCRL, SUCRL is highly sensitive to the knowledge we have on the distribution of $R_{\mathcal{O}}$ and $\tau_{\mathcal{O}}$. In particular, if our knowledge on $R_{\mathcal{O}}$ and $\tau_{\mathcal{O}}$ is very loose, SUCRL can even perform worse than UCRL for all values of $T_{\max}$. Although we expect SUCRL to perform better than FSUCRL due to the additional knowledge provided to the algorithm, the fact that FSUCRL may perform better than UCRL can be explained as follows. FSUCRL not only exploits correlations between options sharing state-action pairs (by collecting samples at action level and not at option level like SUCRL), but it also leverages over the correlation between $R_{\mathcal{O}}$ and $\tau_{\mathcal{O}}$ within a single option (by being optimistic on the ratio $\overline{R}_{\mathcal{O}}/\overline{\tau}_{\mathcal{O}}$ directly through the stationary distribution instead of $\overline{R}_{\mathcal{O}}$ and $\overline{\tau}_{\mathcal{O}}$ separately as in SUCRL).

### F.2 Four-rooms maze

We now consider the famous four-rooms environment introduced in [1]. This domain is characterized by having four cardinal actions with a probability $0.2$ of failure (uniformly in any other direction). The grid-world is a square of dimension 14x14 (for short $d = 14$) with every room being a square of

dimension 7x7. Each room has exactly two exit doors. In every state of every room, we define four options: two are leading to the two exit doors, one is leading to the center of the room, and the last one leads to the unique corner of the grid in the room. Thus, the number of state-options is slightly bigger than the number of state-actions. The optimal policy takes the shortest path to the target state which is located in one of the 4 corners of the grid and the rewards are the same as in the previous experiment. Once the target is reached, the next state is chosen uniformly at random in the grid.

Like in the previous experiments, we ran our algorithms with different configurations summarized in Tab. 3. The configurations replicate the settings presented for the grid-world experiment with the addition of experiments with Bernstein confidence bounds (see Eq. (6a)–(6c)). On Fig. 7, we plot the regret $\Delta(\mathfrak{A}, n)$ as a function of $T_n$ for $\mathfrak{A} \in \{\text{UCRL}, \text{SUCRL}, \text{FSUCRL}\}$. The two versions of SUCRL are exactly the same as in the previous experiments: SUCRLv2 uses $\max_o\{\sigma_\tau(o)\}$ while SUCRLv3 uses $(\sigma_\tau(o))_{o \in \mathcal{O}}$. Note that the other SUCRL versions are not valid in this domain.

**Hoeffding settings.**    We start considering the results provided in Fig. 7a where a small 6x6 grid is considered. Both SUCRL and FSUCRL are outperformed by UCRL by a big margin. The explanation of this negative result resides in the fact that the options deteriorate the navigability of the grid and do not provide any temporal abstraction in such a small domain. Although the reader may not be surprised by this result, we have decided to show it in order to stress the fact that options require a careful design to provide a positive contribution to the learning process.

When we consider bigger mazes the utility of options becomes clear. Fig. 7c (configuration Room-C1) shows the regret achieved by the algorithms when the dominant term is the estimation of the transition kernel ($\Delta_p$). As in the grid-world experiment, FSUCRLv2 performs (on average) similarly to SUCRL with full information (v3) showing that the estimation of option characteristics has a small impact on the overall performance. However, it has a much bigger variance than UCRL or SUCRL. We think the reason comes from the dependency of the regret to the smallest number of visits among all inner states of any option $o$: $1/\min_{s'}\{N_o^k(s')\}$ (see App. D). If all options were uncorrelated, $\min_{s'}\{N_o^k(s')\}$ would behave has $\min_{s'}\{\mu_{s,o}(s')\} \cdot T_k(s, o)$ and so the second term of the regret (i.e., the term not depending on the diameter but on the characteristics of the options) would scale as $1/\mu^*$ where $\mu^* = \min_{s,o}\{\min_{s'} \mu_{s,o}(s')\}$. In our experiments, $\mu^*$ is of the order of $10^{-10}$ and since we chose $T_n = 2 \cdot 10^9 < \mu^*$ we should in theory observe a linear regret. But as we can see on the chart, the regret is far from being linear. We empirically observed that this is due to the fact that options are correlated: for any option $o$, there are inner states that are very unlikely to be visited, but there exist other options where at least one of these states is visited with high probability[19]. All options that have different starting states but identical policies are very much correlated (same inner states and actions). What happens in practice is that FSUCRLv2 relies on these correlations to explore the environment efficiently and avoid scaling with $1/\mu^*$. Notice that correlations are only exploited to compute the characteristics of options and the associated confidence bounds, but when the optimistic policy is computed every option is treated independently. This means that the exploration of an option in the early stages is random since it depends on how well other options are explored. In other words, the optimistic step (EVI), by ignoring correlations, is not able to detect that exploring option $o$ could give insights on the exploration of an option $o'$.

When we decrease $\alpha_p$ and $\alpha_{mc}$ we do not observe any more such a high variance (refer to configuration Room-C2 in Fig. 7d). Recall that by decreasing these factors, the regret term $\Delta_p$ becomes negligible compared to the term $\Delta_{R,\tau}$ depending on the characteristics of the options (reward and holding time). Empirically, we have observed that this variance reduction is mainly due to the decrease of $\alpha_{mc}$ that allows of quickly learning the inner dynamics of options, which results in an overall decrease of the uncertainty (compare Fig. 7c with Fig. 7e). Configuration Room-C2 confirms that SUCRL family suffers the most in this settings due to the black-box view of options (refer to paragraph Grid-C2).

**Bernstein settings.**    By observing Fig 7b (configuration Room-C3) we notice similarities with configuration Room-C2 ($\Delta_{R,\tau}$ is the dominating term due to very small $\alpha_p$ and $\alpha_{mc}$). However, the shrinking coefficients in Room-C3 are more than 10 times bigger. The similar behaviour is explained

| Name | $d$ | $\alpha_p$ | $\alpha_{mc}$ | $\alpha_r$ | $\alpha_\tau$ | Bound | $T_n$ |
|---|---|---|---|---|---|---|---|
| Room-C0 | 6 | 0.2 | 0.2 | 0.8 | 0.8 | Hoeffding | $6 \cdot 10^7$ |
| Room-C1 | 14 | 0.2 | 0.2 | | | Hoeffding | $2 \cdot 10^9$ |
| Room-C2 | 14 | 0.02 | 0.02 | 0.8 | 0.8 | Hoeffding | $1 \cdot 10^9$ |
| Room-C3 | 14 | 0.8 | 0.8 | | | Bernstein | $1 \cdot 10^9$ |
| Room-C4 | 14 | 0.2 | 0.02 | | | Hoeffding | $2 \cdot 10^9$ |

Table 3: Settings used for the four-room maze.

| | Grid-world | | | Four-rooms maze | |
|---|---|---|---|---|---|
| | $T_{\max}$ | | | $d$ | |
| | 5 | 11 | 18 | 6 | 14 |
| $\mu^*$ | 0.007 | 0.015 | 0.006 | $1.5 \cdot 10^{-7}$ | $7.3 \cdot 10^{-10}$ |
| $\kappa^*$ | 1.35 | 1.99 | 2.37 | 3.56 | 12.21 |

Table 4: Examples of options' characteristics for the grid-world and four-room maze.

by the fact that Bernstein bound is tighter than Hoeffding one. It allows to learn the dynamics (SMDP and options transition kernel) before the reward is learned (it exploits Hoeffding bound). As a consequence FSUCRLv2 is able to outperform on average all the other approaches (but it still suffer from high variance) while SUCRL are suffering from the poor dependence on $\Delta_{R,\tau}$.

**Comments on FSUCRLv1.** While Fig. 7a shows that FSUCRLv1 has a regret similar to the one of v2, FSUCRLv1 is suffering a linear regret in the 14x14 maze where FSUCRLv2 is performing quite good. We have empirically investigated that this behaviour is due to the value of the condition numbers $\kappa$ that are too big, and not due to the fact that FSUCRLv1 is considering the maximum $\ell_1$-norm (FSUCRLv2 is implicitly considering a per-state error). To be sure about this, we run FSUCRLv1 with all condition numbers forced to one. We observed that this algorithm had a behaviour similar to FSUCRLv2 (up to the looser $\| \cdot \|_{\infty,1}$). This simple test suggests that both the variants are equally affected by the effective $\mu^*$ (i.e., $\mu^*$ up to correlations between options).

Note that $\kappa = 1$ means that FSUCRLv1 is only suffering from $\min\{N_o^k\}$ (by construction of the confidence bounds) and since it is behaving similarly to v2 it means that also v2 is suffering from $\min\{N_o^k\}$. This is also related to the fact that v1 and v2 are equivalent as long as an option has a state never visited, i.e., $\min\{N_o^k\} = 0$. This property also explains why there are cases where FSUCRLv2 is matching the bad performance of v1 in the initial phase (see Fig. 7c).

Tab. 4 provides additional support to this considerations. By looking at the values $\kappa^*$ for the grid-world we can notice that the value increases with $T_{\max}$. This explains why the regret of FSUCRLv1 starts increasing for large values of $T_{\max}$ (refer to Fig. 6a). If we cross compare the values between grid-world and four-rooms maze we can notice a similarity between grid-world with $T_{\max} = 18$ and a maze of dimension 6. In these domains the performance of the algorithms (v1 and v2) is comparable. When we move to dimension 14 FSUCRLv1 is doomed to perform much worse than v2 due to the big condition numbers[20].

(a) Room-C0

(b) Room-C3

(c) Room-C1

(d) Room-C2

(e) Room-C4

Figure 7: Evolution of the regret $\Delta(\mathfrak{A}, n)$ as $T_n$ increases. All the configurations are tested over 20 repetitions for which we report minimal, maximal and average regret. In the case of FSUCRLv2 we additionally plot all the 20 curves to better explain the dispersion.