[Reviews · NeurIPS 2017]

Reviewer 1



The paper proposes variants of SMDP-UCRL (SUCRL), FSUCRLv1 and v2, and analyzes the regret behavior of the two variants. The crucial difference is that, while SUCRL requires knowledge of certain parameters related to the option behavior, the proposed variants estimate them from the data. The two variants differ in whether the parameters and their confidence intervals are explicitly estimated, or if a "nested" procedure is used to obtain the parameters implicitly. The regret bounds given almost match that of SUCRL, with the exception of some cost for the parameter estimation. Some empirical validation is also presented. The paper is clearly written for most parts. Given the nature of the contribution, it is to be expected that the paper will be dense at some points. The crucial insight is the use of the irreducible Markov chain representation for an option. Having said that, since the primary contribution of the work is Theorem 1, it would be nice to seem intuition for the proof in the main body of the paper. Currently all of the proof is in the appendix.

Reviewer 2



Overview: The authors attempt to improve current regret estimation for HRL methods using options. In particular they attempt to do so in the absence of a distribution of cumulative rewards and of option durations, which is a requirement of previous methods (UCRL and SUCRL). After assuming that options are well defined, the authors proceed to transform the inner MDP of options, represented by a transition matrix Po, into an equivalent irreducible Markov chain with matrix P'o. This is done by merging the terminal states to the initial state. By doing so, and assuming that any state with a termination probability lower than one can be reached, the stationary distribution mu_o of the chain is obtainable; which in turn is utilized to estimate the optimistic reward gain. With respect to previous methods, this formulation is more robust to ill-defined estimates of the parameters, and better accounts for the correlation between cumulative reward and duration. This method is coined as FSUCRL. The algorithm is complemented by estimating confidence intervals for the reward r, for the SMDP transition probabilities, and for the inner Markov Chain P'o. Two versions of this same algorithm, FUSCRL Lvl1 and Lvl2, are proposed. The first version requires directly computing an approximated distribution mu_o from the estimate of P'o. The second version nests two empirical value iterations to obtain the optimal bias. The paper concludes with a theoretical analysis and a numerical application of FSUCLR. On a theoretical ground, FSUCRL Lvl2 is compared to SUCRL. The authors argue that the goodness of the bound on the regret predicted by FSUCRL Lvl2 compared to that of SUCRL depends various factors, including the length of the options and the accessibility of states within the option itself, and provide conditions where FSUCRL Lvl2 is likely to perform better than SUCRL. As an indication, 4 algorithms, UCRL, SUCRL, FSUCRL Lvl1 and FSUCRL Lvl 2 are tested on a gridworld taken from ref. 9, where the maximum duration of the options is assumed known. Result confirm the theoretical expectations previously discussed: FSUCRL Lvl2 performs better of both SUCRL and FSUCRL Lvl1, partially due to the fact that the options' actions overlap. Evaluation: - Quality: The paper appears to be theoretically sound and the problem discussion is complete. The authors discuss in depth strength and weaknesses of their approach with respect to the previous SUCRL. The provided numerical simulation is not conclusive but supports the above considerations; - Clarity: the paper could be clearer but is sufficiently clear. The authors provide an example and a theoretical discussion which help understanding the mathematical framework; - Originality: the work seems to be sufficiently original with respect to its predecessor (SUCRL) and with respect to other published works in NIPS; - Significance: the motivation of the paper is clear and relevant since it addresses a significant limitation of previous methods; Other comments: - Line 140: here the first column of Qo is replaced by vo to form P'o, so that the first state is not reachable anymore but from a terminating state. I assume that either Ass.1 (finite length of an option) or Ass. 2 (the starting state is a terminal state) clarify this choice. In the event this is the case, the authors should mention the connection between the two; - Line 283: "four" -> "for"; - Line 284: "where" s-> "were";

Reviewer 3



This paper provides an extension to Fruit & Lazaric, 2017 for the problem of exploration and exploitation with temporal abstraction in the framework of options. It weakens the previous assumptions on the structure of the given set of options: it does not require inclusion of primitive options or upper bounds on the distributions of the sojourn time or cumulative return. A novelty of the proposed analysis is that it leverages the MDP structure "within" options rather than only the SMDP-level. This perspective is in the spirit of the original options paper "between mdps and smdps [...]" which advocated for a framework capable of interleaving various time scales (mdp or smdp). The analysis is constructed on the observation that the internal structure of each option is amenable to an MDP with a random horizon specified by the termination function. The authors show how an optimistic policy can be written in terms of the "internal" stationary distributions induced by each each option. Confidence bounds are then obtained over these stationary distributions rather than the cumulative return and sojourn times as in Fruit & Lazaric 2017. The theoretical guarantees obtained under this setting capture some interesting properties which have been observed empirically, eg.: long options are harder to learn, options whose transition models have sparser support facilitates learning. Despite its technicality, the main paper remains readable and the main results on the regret are accessible. It should be useful to readers of different backgrounds. ---- - The average reward criterion is adopted throughout the paper. I would like to know if your analysis could also accommodate the discounted setting (which is also more commonly used in practice). - While the appendix motivates the assumption on the initiation sets (only one state per option) it does not address the question of compositionality in Assumption 1. How can the compositionality be satisfied in general ? Line 251 : "[...] somewhat worst-case w.r.t. to the correlation". This statement is unclear. However, the question of correlation between options is interesting and should be developed in the text.